# Brain functional networks associated with social bonding in monogamous voles

M Fernanda López-Gutiérrez[1], Zeus Gracia-Tabuenca[1], Juan J Ortiz[1], Francisco J Camacho[1], Larry J Young[2], Raúl G Paredes[1,3], Néstor F Díaz[4]*, Wendy Portillo[1]*, Sarael Alcauter[1]*

[1]Instituto de Neurobiología, Universidad Nacional Autónoma de México, Querétaro, Mexico; [2]Silvio O Conte Center for Oxytocin and Social Cognition, Center for Translational Social Neuroscience, Yerkes National Primate Research Center, Department of Psychiatry and Behavioral Sciences, Emory University, Atlanta, United States; [3]Escuela Nacional de Estudios Superiores, Unidad Juriquilla, Universidad Nacional Autónoma de México, Querétaro, Mexico; [4]Instituto Nacional de Perinatología Isidro Espinosa de los Reyes, Ciudad de México, Mexico

**Abstract** Previous studies have related pair-bonding in *Microtus ochrogaster*, the prairie vole, with plastic changes in several brain regions. However, the interactions between these socially relevant regions have yet to be described. In this study, we used resting-state magnetic resonance imaging to explore bonding behaviors and functional connectivity of brain regions previously associated with pair-bonding. Thirty-two male and female prairie voles were scanned at baseline, 24 hr, and 2 weeks after the onset of cohabitation. By using network-based statistics, we identified that the functional connectivity of a corticostriatal network predicted the onset of affiliative behavior, while another predicted the amount of social interaction during a partner preference test. Furthermore, a network with significant changes in time was revealed, also showing associations with the level of partner preference. Overall, our findings revealed the association between network-level functional connectivity changes and social bonding.

**\*For correspondence:**
nfdiaz00@yahoo.com.mx (NF);
portillo@unam.mx (WP);
alcauter@inb.unam.mx (SA)

**Competing interests:** The authors declare that no competing interests exist.

## Introduction

The prairie vole (*Microtus ochrogaster*) is a rodent native of North America whose natural behavior involves pair-bonding, which can be defined as a long-lasting, strong social relationship between individuals in a breeding pair in monogamous species (*Walum and Young, 2018*). Pair-bonded voles will usually display selective aggression towards unfamiliar conspecifics; biparental care, including paternal behavior and alloparenting; incest avoidance; and reproductive suppression of adult individuals within a family group (*Carter et al., 1995*). These behaviors make the prairie vole a valuable model to investigate behaviors associated with a socially monogamous reproductive strategy (*Young and Wang, 2004*), social bond disruption, social isolation, and social buffering (*Lieberwirth and Wang, 2016*). Being comparable to human-like social interactions, studying the neurobiology of social behavior in the prairie vole may allow further understanding of human social bonding, its alterations in psychological disorders, and overall impact on health (*Kiecolt-Glaser et al., 2010*).

Pair-bonding-related behaviors in the prairie vole depend on hormonal mechanisms and the activation of emotional, reward, and sensory brain circuits (*Walum and Young, 2018*), integrating large functional networks (*Johnson and Young, 2017*). Among these, the mesolimbic reward system and the social decision-making network (SDMN) (*Johnson et al., 2017*) are the proposed networks to be involved in pair-bonding, being modulated by steroid hormones, dopamine (DA), oxytocin (OXT), arginine vasopressin (AVP), γ-aminobutyric acid, glutamate, and corticotropin-releasing factor,

among others (*Walum and Young, 2018*). A current hypothesis suggests that pair-bonding consists of two different plastic processes: the formation of a neural representation of the partner, and a selective preference for the partner, that is the maintenance of the pair-bond (*Young and Wang, 2004*; *Walum and Young, 2018*). In this process, an association has to be created between the rein-forcing properties of sex (mating) and the olfactory signature from the partner (*Ulloa et al., 2018*; *Young and Wang, 2004*). In broad terms, both OXT and AVP are necessary and sufficient for the formation of the pair-bond (*Young and Wang, 2004*; *Lieberwirth and Wang, 2016*), and their release during social and sexual interactions are the likely triggers and critical modulators of the mentioned network, since most regions included in the SDMN express their corresponding receptor binding (*Johnson and Young, 2017*). DA would also be released in concert with OXT and AVP in specific regions to modulate the adequate display of behavior and formation of the pair-bond (*Young and Wang, 2004*).

While prairie voles of both sexes already display changes in behavior by 24 hr of cohabitation with mating as a result of pair-bonding (*Wang and Aragona, 2004*; *Williams et al., 1992*), the nucleus accumbens (NAcc), for example, has substantially more D1-like receptor binding 2 weeks after female exposure, relative to non-pair-bonded males (*Aragona et al., 2006*), a phenomenon that is also been observed in female voles (*Resendez et al., 2016*). This evidence suggests that long-term plasticity is relevant for pair-bond maintenance. Furthermore, male prairie voles that were pair-bonded for 2 weeks displayed selective aggression toward conspecific male and female strang-ers (*Gobrogge et al., 2007*), and the 2 week time frame has also been found relevant in bond disruption (*Insel and Hulihan, 1995*) and partner loss (*Tabbaa et al., 2017*). While there is a possibility that these changes in behavior involve the interplay of broad networks, it has not been directly explored in vivo.

Recently, novel electrophysiologic and optogenetic techniques have been employed to demon-strate that the functional connectivity between the NAcc and the medial prefrontal cortex (mPFC) during initial cohabitation in female prairie voles modulates the affiliative behavior with their poten-tial partner (*Amadei et al., 2017*), providing exciting data of the relevance of such corticostriatal interactions for social bonding. However, this approach does not allow the study of the interaction of multiple brain regions, that is networks, and their relevance of such interactions in the process of pair-bonding. Neuroimaging methods may provide the alternative to explore such networks in a lon-gitudinal fashion, since few studies have made use of positron-emission tomography to explore lim-ited aspects of such longitudinal changes (*Bales et al., 2007*), providing the first longitudinal evidence of neurophysiological changes associated with pair-bonding. Potentially, functional mag-netic resonance imaging (fMRI) may be the ideal tool to explore the longitudinal changes in func-tional brain networks (*Damoiseaux et al., 2006*), providing high spatial resolution, wide brain coverage, and being minimally invasive to explore longitudinal changes. In particular, resting-state functional magnetic resonance imaging (rsfMRI) explores the low-frequency fluctuations (<0.1 Hz) of the blood-oxygen-level-dependent (BOLD) signal, which has proven to remain highly synchronized within the sensory, motor, and associative networks in the brain of humans (*Damoiseaux et al., 2006*), non-human primates (*Rilling et al., 2007*), and rodents (*Gozzi and Schwarz, 2016*), including the prairie vole (*Ortiz et al., 2018*), turning this technique into a promising tool for translational research. Indeed, functional connectivity explored by means of the correlation of rsfMRI signals of anatomically separated brain regions (*Friston et al., 1993*) has demonstrated to be correlated with neuronal activity (*Mateo et al., 2017*), and it is subject to change in plastic processes such as learn-ing and memory in both humans (*Jolles et al., 2013*) and rodents (*Nasrallah et al., 2016*). A recent study found differences in diffusion-weighted imaging and resting-state functional connectivity between two distinct populations of male prairie voles (*Ortiz et al., 2020*). Here, we make use of this non-invasive technique to explore behavioral correlates with brain functional connectivity and its longitudinal changes associated with pair-bonding in male and female prairie voles.

## Results

### Baseline functional connectivity predicts the display of affiliative behavior

Thirty-two 3-month-old sexually naïve female (N = 16) and male (N = 16) prairie voles (*M. ochrogaster*) were used in the study. Prairie voles underwent three magnetic resonance imaging (MRI) acquisition sessions: a baseline scan before cohabitation, a second scan 24 hr after the onset of cohabitation, and a third scan 2 weeks after the onset of cohabitation (*Figure 1A*). The final imaging sample consisted of 90 datasets, with only six subjects missing one session (see Materials and methods). The day after the baseline scanning session, female and male voles unrelated to each other were randomly assigned as couples and placed for cohabitation in a new home to promote ad libitum mating and social interaction. Four days before cohabitation, silastic capsules (Dow Corning

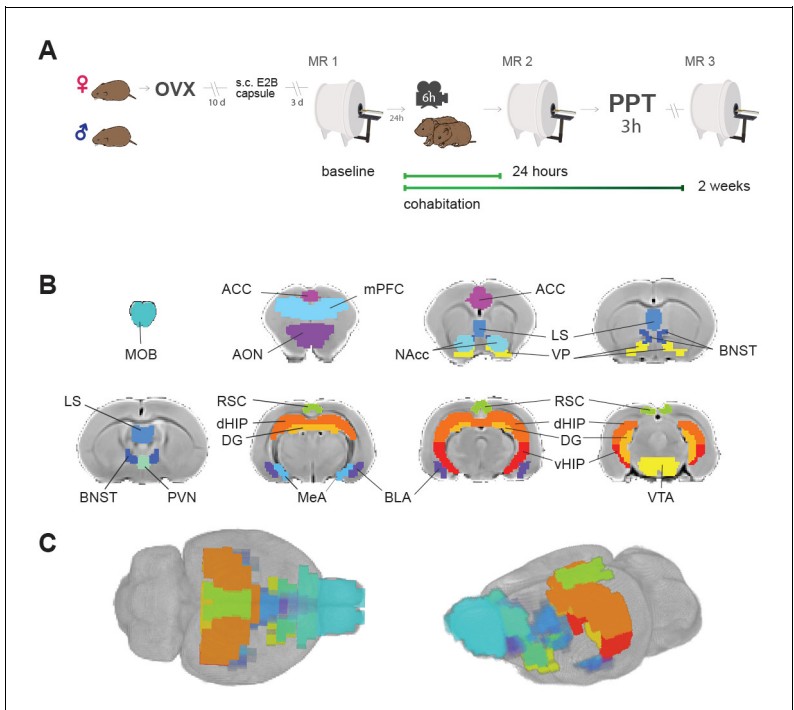

**Figure 1.** Experimental design and brain regions of interest. (**A**) Sequence of experiments during a 30 day period: Female voles were bilaterally ovariectomized before MR and behavioral protocols. After being allowed to recover from surgery for 10 days, silastic capsules containing E2B (estradiol benzoate) were implanted via s.c. 4 days before cohabitation for sexual receptivity induction. Once couples went under cohabitation, they were housed together for the rest of the experiment and were only separated for PPT and MR scanning sessions. OVX: ovariectomy surgery. MR: magnetic resonance imaging scanning session. PPT: partner preference test. (**B**) Regions of interest (ROIs) for network functional connectivity analyses. Antero-posterior coronal slices of the prairie vole template overlayed with ROI masks with the resolution used in the analysis. Each color represents a different ROI. ACC: anterior cingulate cortex. AON: anterior olfactory nucleus. BLA: basolateral amygdala. BNST: bed nucleus of the stria terminalis. DG: dentate gyrus. dHIP: dorsal hippocampus. MeA: medial amygdala. MOB: main olfactory bulb. LS: lateral septum. mPFC: medial prefrontal cortex. NAcc: nucleus accumbens. PVN: paraventricular nucleus. RSC: retrosplenial cortex; VP: ventral pallidum. vHIP: ventral hippocampus. VTA: ventral tegmental area. (**C**) 3D views of ROI masks embedded within the prairie vole template.

The online version of this article includes the following figure supplement(s) for figure 1:

**Figure supplement 1.** Representative rsfMRI time series.

**Figure supplement 2.** Average functional connectivity correlation matrices between ROIs in all subjects shown by MR acquisition sessions: baseline, 24 hr, and 2 weeks.

**Figure supplement 3.** Representative rsfMRI and anatomical raw data and examples of the registration steps.

**Figure supplement 4.** Average seed-based functional connectivity maps for each of the 16 ROIs here explored.

**Figure supplement 5.** Vole temperature during acquisition, discarded images and signal-to-noise ratio.

Silastic Laboratory Tubing; ThermoFisher Scientific, Pittsburg, PA) containing estradiol benzoate (E2B; Sigma–Aldrich, St. Louis, MO) were implanted in previously ovariectomized female voles to enable sexual receptivity and promote mating (see Materials and methods). The first 6 hr of cohabitation was video recorded for analysis of social and mating behavior. Mount (M ± SEM: 65.4 ± 31.7 min), intromission (116 ± 35.3 min), and ejaculation (125 ± 34.4 min) latencies were obtained for male voles (N = 16). Lordosis latency (22.3 ± 13.3 min) was also measured on females (N = 16), and huddling latencies (69.5 ± 15.8 min) were obtained for each male and female pair. Three of the 16 couples did not mate during the recorded period, but all voles displayed huddling and licking/grooming behavior with their sexual partner. Once joined, voles remained housed in couples for the rest of the experiment and were only separated for MRI scanning sessions and behavioral tests.

To explore relationships between socio-sexual behavior and functional connectivity, 16 regions of interest (ROIs) were defined according to their previously reported relevance in the process of pair-bond formation and maintenance (*Johnson and Young, 2017*; *Lieberwirth and Wang, 2016*; *Walum and Young, 2018*), which were the following: anterior cingulate cortex (ACC), anterior olfactory nucleus (AON), basolateral amygdala (BLA), bed nucleus of the stria terminalis (BNST), lateral septum (LS), medial amygdala (MeA), main olfactory bulb (MOB), medial prefrontal cortex (mPFC), nucleus accumbens (NAcc), retrosplenial cortex (RSC), paraventricular nucleus of the hypothalamus (PVN), ventral pallidum (VP), ventral tegmental area (VTA), dentate gyrus (DG), dorsal hippocampus (dHIP), and ventral hippocampus (vHIP) (*Figure 1B,C*). For each subject and session, average time series for each region were extracted from the pre-processed fMRI datasets (*Figure 1—figure supplement 1*). The latter were used to obtain connectivity matrices, based on the partial-correlation estimates for all possible pairs of ROIs (*Figure 1—figure supplement 2*).

The amount and proneness to affiliative behavior has shown reliability in describing the level of sociability and mating systems in Microtus voles. Through the measurement of huddling, *Salo et al., 1993* were able to distinguish social behavior in four different species of voles by scoring huddling when the pair was either sitting or lying in bodily contact, with the prairie vole having the highest huddling accumulation. Other studies in voles have also measured huddling latency to assess levels of affiliative behavior, since it may influence pair-bond induction and formation (*Shapiro and Dewsbury, 1990*; *Amadei et al., 2017*).

In order to identify potential relationships between behavior and functional connectivity with a network perspective, huddling latencies during cohabitation in voles of both sexes (N = 28) were tested as linear covariates of baseline functional connectivity data, that is before cohabitation, with Network-Based Statistic (NBS) framework, using the NBS Toolbox (*Zalesky et al., 2010*).

NBS analysis found significant negative linear relationship ($p \leq 0.001$) with a large network including VP, MeA, LS, VTA, RSC, BLA, NAcc, ACC, and the DG (*Figure 2A*). Our results show that the higher the connectivity between these regions before cohabitation, the shorter the huddling latencies during cohabitation in voles of both sexes. Additionally, a posteriori Pearson's correlations confirmed the correlation strength between each connection and huddling latencies: VP–MeA (r(26) = −0.468, p=0.011), MeA–LS (r(26) = −0.372, p=0.051), LS–VTA (r(26) = −0.502, p=0.006), VTA–RSC (r(26) = −0.586, p=0.001), RSC–PVN (r(26) = −0.382, p=0.044), RSC–BLA (r(26) = −0.362, p=0.058), BLA-DG (r(26) = −0.420, p=0.025), BLA–NAcc (r(26) = −0.543, p=0.002), and NAcc–ACC (r(26) = −0.378, p=0.047), (*Figure 2B–J*). These results show that functional connectivity between these regions reflect the predisposition to display affiliative behavior in both female and male prairie voles.

## Prairie voles of both sexes show partner preference after cohabitation

Between 48 and 72 hr of cohabitation, partner preference was evaluated on each subject (N = 32) to assess pair-bonding behavior. This protocol was based on a previously described test (*Williams et al., 1992*; *Figure 3B*, see Materials and methods). The partner preference index revealed a significant difference between the proportion of time spent on the incentive area related to the partner (median = 0.63) with the area related to the stranger vole (median = 0.37) for all subjects (U = 378, p=0.0365, effect size r = 0.32) (*Figure 3B*). No significant differences were found between males and females in their preference for the partner (U = 121, p=0.81, effect size *r* = 0.05) or the stranger voles (U = 118, p=0.72, effect size r = 0.07), and there were also no significant differences in partner preference between the time periods when PPT tests were performed (48 and 72 hr) (U = 119, p=0.75, effect size r = 0.06) (*Figure 3—figure supplement 1*).

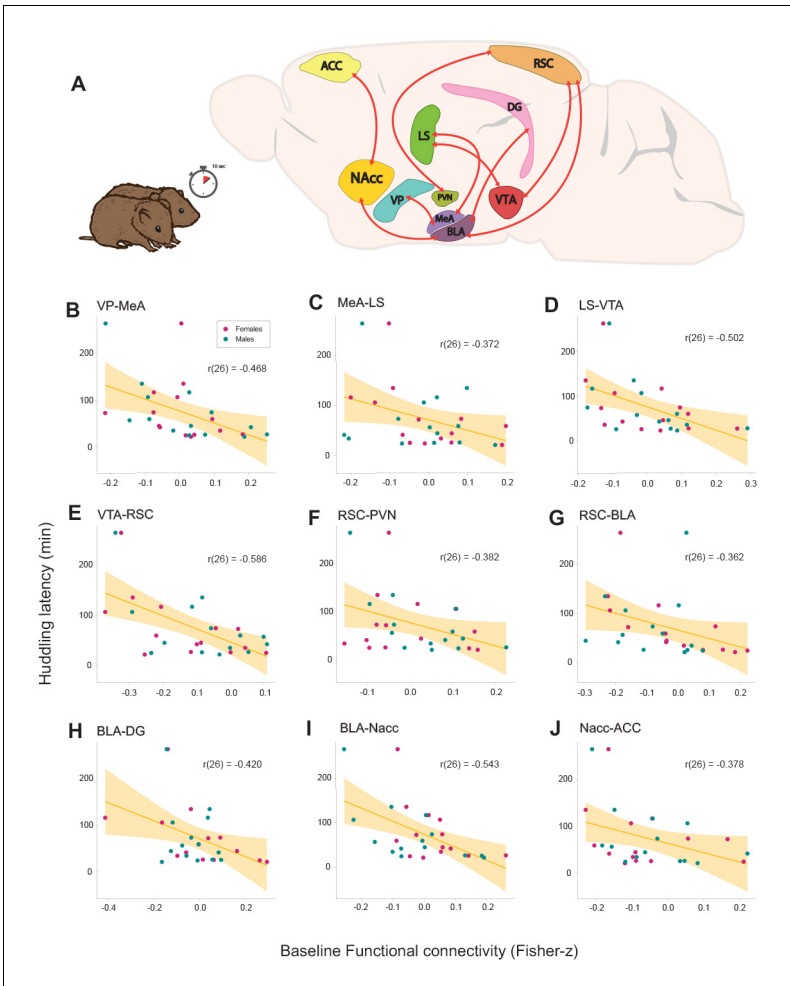

**Figure 2.** Relationships between baseline functional connectivity and affiliative behavior (huddling) during cohabitation with mating in male and female prairie voles. (**A**) Representation of a prairie vole brain with regions (nodes) that constitute the network with a significant negative association with huddling latency. Scatter-plot graphs (**B–J**) of the connections in a with best line fit between baseline functional connectivity (Fisher z-transformed partial-correlation values) and huddling latencies (minutes) during cohabitation. The higher the connectivity between these regions before cohabitation, the shorter the huddling latencies during cohabitation in voles of both sexes. ACC: anterior cingulate cortex. BLA: basolateral amygdala. DG: dentate gyrus. LS: lateral septum. MeA: medial amygdala. NAcc: nucleus accumbens. PVN: paraventricular nucleus. RSC: retrosplenial cortex. VP: ventral pallidum. VTA: ventral tegmental area.

The online version of this article includes the following source data for figure 2:

**Source data 1.** Functional connectivity values of edges that correlate significantly with huddling latencies in prairie voles.

## Functional connectivity at 24 hr of cohabitation predicted social interaction during the partner preference test

During the partner preference test (PPT), the proportion of time spent on social incentive areas over the total recorded time was calculated for analysis in all subjects (see Materials and methods). The percentage of time spent on social incentive areas over the total time of the test had a mean of 79.70 ± 2.65%. Spearman correlation analyses showed that the amount of time spent on social incentive areas had no relationship with the partner preference index (rs(30) = 0.08912 p=0.6277) (*Figure 4—figure supplement 1*). However, we tested if functional connectivity 24 hr after the start of cohabitation had a relationship with the total time spent on social incentive areas during the test, regardless if the subject interacted with the partner or stranger stimulus vole (N = 32). Through NBS

**Figure 3.** Partner preference test. (**A**) Representative figure showing the design of the arena in which voles were tested for partner preference test. (**B**) Between 48 and 72 hr of cohabitation, partner preference was evaluated on each subject (N = 32). Partner preference index revealed a significant difference between the time spent on the incentive area related to the partner, with the incentive area related to the stranger vole. Boxplot graphs show whiskers with 10–90 percentiles; horizontal line inside the box shows data median, and '+' represents data mean. (*) denotes significance at p<0.05.

The online version of this article includes the following source data and figure supplement(s) for figure 3:

**Source data 1.** Values of parter preference index obtained from the proportion of time spent on social incentive areas related to the partner and stranger voles.

**Figure supplement 1.** Partner preference indexes at two different time points.

**Figure supplement 1—source data 1.** Partner preference index comparison at different PPT test times (48 vs 72 hr) from the onset of cohabitation.

analysis, a significant positive linear relationship (p=0.013) was found with the following network: LS–NAcc–mPFC–MeA–VP–MOB–DG (*Figure 4A*). These findings suggest that the lower the connectivity between these regions 24 hr after the onset of cohabitation, the longer the subject would interact with a conspecific of the opposite sex during the PPT, which was evaluated between 48 and 72 hr after the onset of cohabitation. The correlation strength between each connection of such network and the percentage of time on social incentive areas was obtained with a posteriori Pearson's correlations: LS–NAcc ($r(30) = -0.369$, p=0.037), NAcc–mPFC ($r(30) = -0.312$, p=0.081), mPFC–MeA ($r(30) = -0.373$, p=0.034), MeA–VP ($r(30) = -0.376$, p=0.033), VP–MOB ($r(30) = -0.531$, p=0.001), and MOB–DG ($r(30) = -0.404$, p=0.021) (*Figure 4B–G*).

## Male and female voles share network-level changes related to social bonding in different time points

To determine whether cohabitation with mating induced changes in brain functional connectivity between sessions, longitudinal data was analyzed with linear mixed models (LMM), implemented via the Network-Based R-statistics package (NBR; *Gracia-Tabuenca and Alcauter, 2020*), which allows to implement LMM in an NBS framework (see Materials and methods). NBR analysis yielded significant session effects, that is changes between baseline, 24 hr, and 2 weeks of cohabitation ($p_{FWE} = 0.0482$), in a network consisting of 10 regions: ACC, BLA, dHIP, LS, mPFC, NAcc, RSC, vHIP, VP, and VTA. No significant differences were found between male and female voles, a result that led us to run the analysis without the sex variable. The same network component was obtained ($p_{FWE} = 0.042$) (*Figure 5A*).

Post hoc analyses (false discovery rate [FDR] corrected) identified differential longitudinal changes among rsfMRI sessions. Specifically, four connections had significant changes 24 hr after the onset of cohabitation: ACC–vHIP ($t(56) = -2.766$; p=0.0077), LS–RSC ($t(56) = -3.270$; p=0.0018), and LS–VP ($t(56) = -2.540$; p=0.0138) showed increased functional connectivity, while LS–dHIP ($t(56) = 3.004$; p=0.0039) had decreased connectivity in the same period of time. The edges LS–VP, LS–dHIP, and LS–RSC had no differences between baseline (before cohabitation) and the third session (2 weeks after the onset of cohabitation), suggesting acute plastic changes related to 24 hr of cohabitation; however, LS–mPFC ($t(56) = -2.856$; p=0.0060) and VP–VTA ($t(56) = -2.752$; p=0.0079) have increased connectivity, while ACC–LS ($t(56) = 3.227$; p=0.0020) and NAcc–VTA ($t(56) = 2.970$; p=0.0043) had decreased connectivity between the first and third scanning sessions (baseline vs 2 weeks of cohabitation). In addition, four connections reflected differences between 24 hr and 2

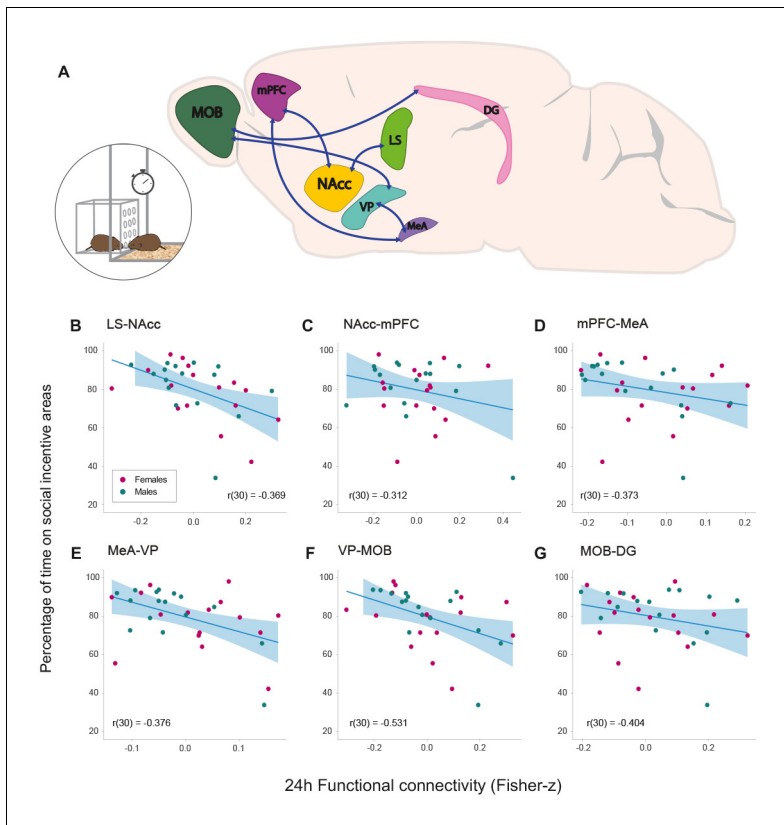

**Figure 4.** Relationships between functional connectivity 24 hr after the onset of cohabitation and social interaction during partner preference test in male and female prairie voles. (**A**) Representation of a prairie vole brain with regions (nodes) that constitute the network with a significant negative association with the amount of social interaction during the PPT. Scatter-plot graphs (**B–G**) of the connections in a with best line fit between baseline functional connectivity (Fisher z-transformed partial-correlation values) and time on social incentive areas during cohabitation (percentage). The lower the connectivity between these regions at 24 hr of cohabitation, the longer the time spent on social incentive areas during the PPT. DG: dentate gyrus. LS: lateral septum. MeA: medial amygdala. MOB: main olfactory bulb. mPFC: medial prefrontal cortex. NAcc: nucleus accumbens. VP: ventral pallidum.

The online version of this article includes the following source data and figure supplement(s) for figure 4:

**Source data 1.** Values of percentage of time spent on social incentive areas related to the partner and stranger voles.
**Figure supplement 1.** Relationships between social behaviors in the prairie vole.

weeks after the onset of cohabitation (sessions 2 and 3): ACC–LS ($t_{(56)}$ = 3.675; p=0.0005) and BLA–NAcc ($t_{(56)}$ = 2.593; p=0.0121) had decreased connectivity, but LS–mPFC ($t_{(56)}$ = −4.013; p=0.0001) and mPFC–dHIP ($t_{(56)}$ = −2.605; p=0.0117) exhibit increased connectivity after 2 weeks of cohabitation (third session), suggesting long-term functional changes related to cohabitation (*Figure 5B–K*). Overall, the node with the most changes in functional connectivity was the LS.

To control for potential confounders, we selected an independent set of brain structures not expected to be involved in social behavior and were tested with the same data and analysis methods than the ROIs mentioned previously. These nine regions were the primary auditory area (AUDp), the cerebellar cortex (CBX), forceps minor of the corpus callosum (fmi), laterodorsal thalamic nucleus (LD), primary motor area (MOp), motor-related medulla (MY), supplemental somatosensory area (SSs), primary visual area (VISp), and ventricle areas (Vent) (*Figure 5—figure supplement 1*). We used the same data and analysis methods to test an independent set of brain structures not expected to be involved in social behavior. Linear mixed models (LMM) implemented via the Network-Based R-statistics package (NBR; *Gracia-Tabuenca and Alcauter, 2020*), considering sex as a fixed variable, and session and intercept as random variables, found no significant networks with

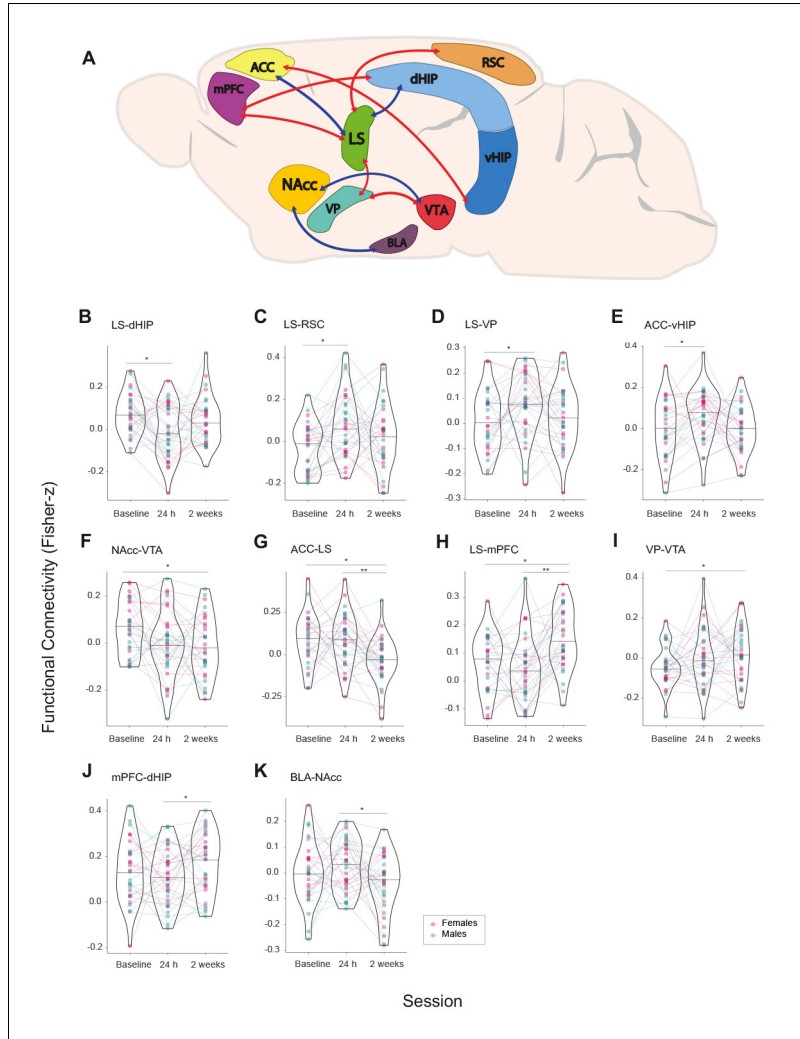

**Figure 5.** Network-wide changes in time in brain functional connectivity of female and male prairie voles. (**A**) NBR analysis via Linear mixed models (LMM) analysis results represented in a prairie vole brain with regions (nodes) comprising the brain network that undergoes significant changes in functional connectivity (Fisher z-transformed correlation values) after cohabitation with mating. Interregional connectivity (edges) is shown by color code. Red: increase of functional connectivity; blue: decrease of functional connectivity. ACC: anterior cingulate cortex. BLA: basolateral amygdala. dHIP: dorsal hippocampus. LS: lateral septum. mPFC: medial prefrontal cortex. NAcc: nucleus accumbens. RSC: retrosplenial cortex. vHIP: ventral hippocampus. VP: ventral pallidum. VTA: ventral tegmental area. (**B–K**) Functional connectivity values in violin plots showing full distribution of data and median. Connecting lines track longitudinal data of each subject between regions through specific MR acquisition time points (Session): Baseline, 24 hr, and 2 weeks of cohabitation. Color codes for data points and connecting lines distinguish male (cyan) from female subjects (pink). False discovery rate (FDR) post hoc significant differences are shown: *<0.05, **<0.01, ***<0.001.

The online version of this article includes the following source data and figure supplement(s) for figure 5:

**Source data 1.** Functional connectivity values that changed significantly across sessions (Ses; 1, 2, and 3) in male and female prairie voles (females: F; males: M).

**Source data 2.** Estimated marginal means (EMM) post hoc comparisons values obtained from the longitudinal analysis of functional connectivity data (edge), via linear mixed models with NBR.

**Figure supplement 1.** Control regions for the longitudinal analysis of rsfMRI data.

**Figure supplement 2.** Ten components gICA.

differences between sessions ($p_{FWE}$ = 0.1568), or sex*session interactions ($p_{FWE}$ = 0.619 and $p_{FWE}$ = 0.3554), suggesting that in both female and male prairie voles, cohabitation with mating and social bonding do not influence changes in functional connectivity in these regions between any session (baseline, 24 hr, and 2 weeks after the onset of cohabitation).

Group-independent component analysis (gICA) was also performed to address the exploration of large-scale functional brain networks and potential differences between sessions. The gICA revealed five components associated with sensory and motor cortices, putative default-mode, and salience networks, a striatum-centered component, and two components with relevant connectivity of the ventral hippocampi, with a degree of lateralization found in some of the latter (*Figure 5—figure supplement 2*). These results are strikingly similar to other networks reported previously in the male prairie vole (*Ortiz et al., 2018*), although this is a larger sample also including female voles and an optimized anesthesia protocol for the detection of rsfMRI networks in rodents (*Grandjean et al., 2020*). In order to evaluate whether networks found through gICA maps had significant changes between sessions, dual regression was applied and group differences across sessions were assessed using two-sample paired t-tests (see Materials and methods). However, the obtained FWE-corrected p statistics found no significant differences between sessions. Being a voxel-wise method, it may not be sensitive enough to detect punctual, region-specific changes in brain functional connectivity.

## Specific network connections that undergo changes during cohabitation with mating correlate with partner preference in male and female voles

Although several regions were found to change significantly after cohabitation and social bonding, we tested if any specific connection of the detected network component (*Figure 5A*) could have a relationship with the partner preference index obtained between 48 and 72 hr after the onset of cohabitation, which was used to evaluate the level of pair-bonding in each subject. Two-tailed Pearson's correlation tests revealed a significant positive relationship in LS–VP baseline functional connectivity (r(26) = 0.435, p=0.020) with the partner preference index, suggesting that the higher the baseline functional connectivity between these regions, the higher the partner preference index would be (*Figure 6A*). Also, significant negative relationships were found between the partner preference index and BLA–NAcc functional connectivity at 24 hr of cohabitation (session 2) (r(30) = −0.464, p=0.0074), and in LS–RSC functional connectivity at 2 weeks of cohabitation (session 3) (r(28) = −0.437, p=0.015), which may suggest that the lower the connectivity between these regions at the specified time points, the higher the partner preference index (*Figure 6B,C*).

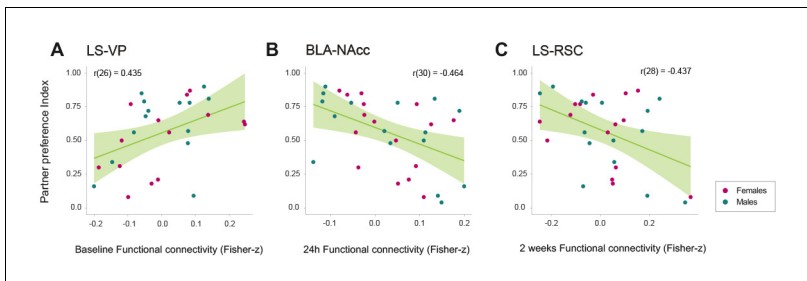

**Figure 6.** Relationships between relevant network connections and partner preference index during cohabitation with mating in male and female prairie voles. Scatterplots (**A–C**) that show significant correlations (with best line fit) between functional connectivity (Fisher z-transformed partial-correlation values) and partner preference index in network connections that undergo longitudinal changes. BLA: basolateral amygdala. LS: lateral septum. NAcc: nucleus accumbens. RSC: retrosplenial cortex. VP: ventral pallidum.

The online version of this article includes the following source data for figure 6:

**Source data 1.** Functional connectivity values of edges that correlate significantly with partner preference index of prairie voles in each session: session 1 (s01), session 2 (s02), and session 3 (s03), which correspond to baseline, 24 hr, and 2 weeks after the onset of cohabitation, respectively.

## Discussion

Several studies have described the relevance of different brain regions involved in pair-bond induction and maintenance in prairie voles (*Johnson and Young, 2017*; *Walum and Young, 2018*). However, longitudinal explorations of the brain before and after pair-bonding are scarce (*Bales et al., 2007*), especially from a network perspective. Here, by using rsfMRI, we were able to detect a brain network in which baseline functional connectivity (before cohabitation) predicted the latency for huddling behavior during the first hours of cohabitation, providing the potential neurofunctional substrate for the variability in affiliative behavior and further extending the recent findings that the corticostriatal electric activity modulates social bonding in prairie voles (*Amadei et al., 2017*). A relationship between functional connectivity and social interaction was also found, in which a network detected from data obtained 24 hr after the onset of cohabitation predicted the amount of social interaction during the PPT. Finally, our results reflected significant longitudinal changes in functional connectivity of prairie voles after pair-bonding. Post hoc analyses revealed differential short- and long-term connectivity changes mainly involving the lateral septum (LS), with three network connections correlating with the level of partner preference in different sessions. We further discuss the potential neurophysiological basis and implications of our findings.

### Correlations between network functional connectivity and social behavior

Although each of the identified networks would likely act as a whole or possess emergent properties, in the following text, we mention previous evidence in rodents that may aid in their functional interpretation, and each component will be dissected in segments to better understand node relationships and their putative role in social bonding and related behavior.

Even though it has been reported that in prairie voles, 24 hr of cohabitation or 6 hr of ad libitum mating is sufficient for a pair-bond to be developed (*Williams et al., 1992*), a considerable amount of evidence has shown that other factors influence its development and maintenance. Specifically, AVP (*Ophir et al., 2008*) and OXT receptor gene expression and density (*King et al., 2016*), paternal nurturing (*Ahern and Young, 2009*), and neonatal isolation (*Barrett et al., 2015*) have shown to produce variability in the exhibition of prairie vole social behavior. Though they were under the same experimental conditions, subjects from both sexes in this study showed a wide behavioral variability not only during their sexual encounters in the first hours of cohabitation, but also in their bonding behavior evaluated between 48 and 72 hr after the onset of cohabitation. It is likely that the sum of previously mentioned factors gives each subject a distinctive brain network configuration that ultimately relates to bonding behavior. Hence, we hypothesized that there may be individual differences in functional connectivity that could explain the variability in socio-sexual behavior.

### Correlation with huddling latencies

Indeed, we identified a network for which the functional connectivity at baseline was negatively related to huddling latencies during the first hours of cohabitation. In other words, baseline functional connectivity predicted how quickly subjects would begin affiliative huddling with an opposite-sex conspecific. Huddling is a measurable affiliative behavior in prairie voles and a useful indicator of social receptiveness (*Salo et al., 1993*). The network included the following connections: VP–MeA–LS–VTA–RSC–PVN, PVN–RSC–BLA–NAcc–ACC–NAcc, and BLA–DG (*Figure 3A*). This network has regions reported to be involved in social salience, social memory and recognition, spatial memory, and reward-seeking mechanisms. Therefore, it is possible that subjects that exhibited huddling at an earlier time found it more rewarding than those who did not.

The VP plays a major role in reward and motivation (*Smith et al., 2009*). Since MeA activity in rodents is necessary for social recognition (*Ferguson et al., 2001*) and responds to sex-specific chemosensory cues (*Wang and Young, 1997*; *Yao et al., 2017*), increased MeA-VP connectivity may improve the rewarding response of sex-related social interaction. Additionally, DA neurons are reported to modulate VP responses evoked by amygdala stimulation (*Maslowski-Cobuzzi and Napier, 1994*). The LS is a structure involved in social recognition and retrieval of relevant social information (*Bielsky et al., 2005*), in addition that it is known to be anatomically interconnected with regions involved in social behavior, including the hippocampus and the MeA (*Risold and Swanson, 1997*). In rats, it has been shown that OXT release in the LS is required for the maintenance of social

memory and may also be modulated by the MeA according to the relevance of the social stimulus (*Lukas et al., 2013*). It is likely that MeA-LS connectivity further integrates social information. LS–VTA–RSC functional connections, through the VTA, may contribute to linking contextual information with the midbrain DA system and regulate motivational behaviors (*Luo et al., 2011*; *Maeda and Mogenson, 1981*), since the RSC is known to be involved in the processing of spatial and contextual memory in rodents (*Todd and Bucci, 2015*).

Connectivity between PVN–RSC–BLA may be related to an association of a spatial context with a socially salient stimuli. While OXT projections from PVN may probably influence the regulation of social salience in the whole network (*Johnson et al., 2017*), it is also reported to be sensitive to external stimuli (*Anacker et al., 2014*; *Liu et al., 2001*) that can impact OXT synthesis and release (*Smith and Wang, 2014*). The BLA may act as an associative site for stimulus-outcome representations (*Cardinal et al., 2002*), which would allow an appropriate response according to previous social encounters, information that possibly requires hippocampus-associated memory (BLA-DG) (*Frey et al., 2001*; *Tashiro et al., 2007*).

In a previous report, the BLA and the ACC were found relevant in coordinating brain activity when social interaction is initiated and in the formation of social recognition memory through gene expression (*Tanimizu et al., 2017*). The NAcc is known to translate reward-predictive information from the amygdala (BLA–NAcc) to promote cue-evoked, reward-seeking behavioral responses in rodents (*Ambroggi et al., 2008*), while the ACC has been considered an important region in the decision-making process between sensory perception, motivation, and final motor performance (*Assadi et al., 2009*). Thus, input from the ACC (NAcc–ACC) may be necessary for a social decision-making process. It is important to note that a recent study demonstrated this particular circuit may be unique in prairie voles (*Horie et al., 2020*), and in female prairie voles, the functional connectivity of the prefrontal cortex and NAcc after the first encounter predicts affiliative huddling toward a partner, and the activation of such circuit biases later preference toward a partner (*Amadei et al., 2017*). Our results extend such findings, showing that a larger network including similar corticostriatal connectivity, measured even before the exposure to a potential partner, predicts affiliative behavior. Moreover, such relation is consistent for both males and females, and the circuit includes the amygdala and hippocampus, as predicted by *Amadei et al., 2017*. Overall, the functional connectivity of this network may indicate the predisposition of a prairie vole to engage in prosocial behavior, which might be influenced by previous experience and other factors mentioned beforehand.

## Correlation with amount of social interaction

A network component detected at 24 hr after the onset of cohabitation, LS–NAcc–mPFC–MeA–VP–MOB–DG, was negatively correlated with the amount of social incentive a subject would have during the partner preference test, a test in which it interacted with two conspecifics of the opposite sex, one being the partner and the other a stranger vole. Interestingly, this component includes five regions from the huddling network previously described, and all but two nodes (MeA–VP) are related differently when comparing it with the former network.

Regardless of the level of preference for the partner, prairie voles of both sexes had a differential interest in engaging socially. While the huddling network is related to pre-pair-bonding affiliative behavior, this network is necessarily related to post-bonding behavior, since it was only detected 24 hr after the onset of cohabitation. It is possible that functional connectivity of this component is associated with the modulation of a behavioral response, that is approach or avoidance, according to social olfactory cues that may be rewarding according to previous experience. Besides its social recognition role, in rodents the LS is reported to modulate reward response by decreasing neuronal activity in the NAcc through AVP release (*Gárate-Pérez et al., 2021*). The NAcc–mPFC pathway has been widely characterized as modulating reward-seeking, goal-oriented behavior (*Gill et al., 2010*), and the mPFC and the amygdala are extensively interconnected and tune the expression of fear and anxiety (*Liu et al., 2020*; *Marek et al., 2013*). The involvement of the MeA–VP hints modulation of sex-related social interaction, particularly via chemosensory stimuli (MOB-DG) (*Castro et al., 2020*; *Liu et al., 2014*).

Our data suggests that functional connectivity 24 hr after the onset of cohabitation has a relationship to the level of sociability, reflecting a phenotype independent of pre-bonding behavior and partner preference (*Figure 4—figure supplement 1*). Further investigation of this network would be necessary to understand the impact of sociability in other behaviors, such as mate-territory guarding

and parental nurturing. Apparently, its change in modulation is on a short term, since a correlation at 2 weeks of cohabitation was not found and other regions or networks may play this role on a long-term basis.

## Longitudinal changes in functional connectivity after pair-bonding

It has been proposed that pair-bonding results from the convergence of the mesolimbic DA reward circuit and social discrimination circuits (**Walum and Young, 2018**). The results here presented are consistent with this model, by demonstrating both short- and long-term changes in a brain network including regions largely associated with reward/motivation (ACC, VTA, NAcc, VP) and the social decision-making network, specifically related to sensory contextualization (BLA, LS, RSC, dHIP), saliency processing (LS, ACC, VTA), and memory formation and retrieval (vHIP, dHIP, mPFC, RSC).

Changes in functional connectivity after 24 hr after the onset of cohabitation may result from familiarization of a new spatial context that also implies a novel social context, that is exposure to new housing and to a novel, opposite-sex conspecific stranger. During this process, subjects are exposed to novel stimuli and engage in their first socio-sexual interactions, forming new memories of the partner and after-bonding behaviors such as mate and territorial guarding may appear. As mentioned earlier in this section, functional connectivity changes detected at 2 weeks after the onset of cohabitation may be related to long-term modulation of behavior as a result of social bonding, in which the partner and its associated cues become salient/rewarding and the pair-bond enters a maintenance phase. In general, these temporally dynamic functional connectivity changes may be related to the modulation of socio-sexual interactions with the partner and add a level of complexity that can be glimpsed due to the longitudinal analysis of the data. While the nodes here identified are interconnected into a larger network and their precise contribution to complex social behavior remains elusive to our methods, their potential role will be discussed based on previous results and the connectivity patterns here identified.

The lateral septum (LS) strikes as a relevant region in the modulation of social behavior, appearing as a hub that may integrate or relay spatial, contextual, and reward information between the ventral striatum and limbic regions, with cortical areas and hippocampal structures (**Wirtshafter and Wilson, 2020**). The hippocampus may also participate as an important relay region, being anatomically connected to the DG, RSC (**Sugar et al., 2011**), VTA (**Gasbarri et al., 1994**), and the prefrontal cortex (**Preston and Eichenbaum, 2013**). Twenty-four hours after the onset of cohabitation, the LS showed a decrease of functional connectivity with the dHIP; in mice, silencing this circuit decreases social aggression (**Leroy et al., 2018**). At the same period, an increase of connectivity between the LS and the RSC was also identified, which may be related to contextual memory acquisition (**Opalka and Wang, 2020**) and possibly associate social information with a spatial context. This change of connectivity may involve relay from the hippocampus, since anatomical projections between them are known (**Risold and Swanson, 1997**). In regard to the VP, it has been reported that its stimulation increases c-fos expression in regions including the LS and could be mediating reward processes (**Panagis et al., 1997**). The increase of connectivity between the latter regions seems to be on a relatively short term.

Social interaction would necessarily require new memory consolidation and, possibly, formation of the neural representation of the partner. Our data showed functional connectivity changes occurred in the hippocampus 24 hr after the onset of cohabitation (ACC–vHIP, LS–dHIP; *Figure 5*), which has demonstrated to be critical in encoding spatial and mnemonic information in rodents (**Lin et al., 2017**). ACC–vHIP activity has been associated with the regulation of fear in novel environments and in contextual fear generalization in rodents (**Bian et al., 2019**). However, the vHIP has also shown to contribute in social memory processing (**Chiang et al., 2018**), and the ACC is proposed to integrate new information to modify future behavior (**Kolling et al., 2016**). Increased ACC–vHIP connectivity could be related to social memory formation, but it may also be related to higher anxiety triggered by a new spatial and social environment.

In regard to functional connectivity changes 2 weeks after the onset of cohabitation, a long-term increase of connectivity was detected between the LS–mPFC, while a decrease was found in ACC–LS connectivity. It is enticing to propose that the LS is necessary for conspecific recognition and subsequent modulation of behavior, since AVP administration enhances pair-bond formation in prairie voles (**Liu et al., 2001**), and male voles exposed to females have increased Fos-immunoreactive cells in the LS (**Wang et al., 1997**). In rats, OXT release in the LS is also required for the maintenance of

social memory, modulated by the relevance of the social stimulus (*Lukas et al., 2013*). Consequently, LS connectivity changes might be related to long-term social recognition and behavior modulation, mediated by afferents from prefrontal structures, that is the mPFC and the ACC (ACC–LS–mPFC). The aforementioned ACC may have a role in saliency regulation and in decision-making processes, while the mPFC is proposed to have a role in the initiation, maintenance, and modulation of social attachment and behavior in rodents (*Ko, 2017*; *Tanimizu et al., 2017*).

We also observed a decrease of connectivity between NAcc and VTA by the 2 week period. Considering the NAcc is densely innervated by VTA DA neurons (*Steinberg et al., 2014*), the after-bonding decrease of connectivity between these two regions may reflect the plasticity events necessary for pair-bond maintenance, in which D1-like receptor upregulation induces a long-term behavioral state in which stranger voles are attacked or avoided (*Aragona et al., 2006*). In mice, VP neuron activity drives positive reinforcement through projections to the VTA (*Faget et al., 2018*). The long-term increase of functional connectivity between these two regions suggests they may play a role in the maintenance of the bond through positive reinforcement.

Functional connectivity between the mPFC and the dHIP was increased. The dorsal region of CA2 has been characterized as a hub for sociocognitive memory processing in rodents, specifically for social memory encoding, consolidation, recall (*Meira et al., 2018*), and social recognition (*Hitti and Siegelbaum, 2014*). Also, the mPFC has been proposed as a critical region for remote memory retrieval and for memory consolidation, reportedly relying on the hippocampus (*Euston et al., 2012*). Therefore, their increased interaction could support the integration of new memories into pre-existing networks (*Preston and Eichenbaum, 2013*) and, potentially, in the incorporation of the partner into long-term memory representations. Indeed, OXT receptor activation in dHIP promotes the persistence of long-term social recognition memory in mice (*Lin et al., 2018*).

Lastly, we detected a decrease of BLA–NAcc connectivity between 24 hr and 2 weeks of cohabitation. It has been proposed that these regions regulate social reward processing in rats (*Ambroggi et al., 2008*) and may facilitate pair-bonding in prairie voles (*Horie et al., 2020*). Our results suggest that these regions may encode the reward-predictive cue, that is the partner, during bond establishment, but other regions could be involved in the maintenance of the bond, such as VP–VTA and NAcc–VTA.

Although both AVP and OXT have been proposed to have sex-differentiated roles in pair-bond formation (*Young and Wang, 2004*), the functional connectivity here explored showed no significant differences between sexes or significant sex-session interactions. Sexual dimorphism may still be present in a structural level, but the outcome of brain functional connectivity could remain similar between sexes.

Even though our results show ample individual variability in display of behavior as much as in brain functional connectivity, our analysis was able to capture group-wise consistent changes for both female and male prairie voles, suggesting these could be crucial in modulating social behavior after cohabitation with mating independently of the strength of the pair-bond. We propose that the long-term functional connectivity changes observed in the network are related to social bonding and may lead to pair-bonding induction and maintenance.

Concerning gICA, the obtained FWE-corrected P statistics found no significant differences between sessions. This being a voxel-wise type of analysis that implies multiple comparisons with its respective correction, gICA maps may not be sensitive enough to detect punctual, region-specific changes in brain functional connectivity. While ICA in rsfMRI is a valuable method with advantageous properties for identifying spatio-temporal signal sources that are not well-characterized or well-understood, different methods of analyses such as NBS seem more suitable for the aim of our study, which was to investigate the relationship between social behavior and functional connectivity, as well as the exploration of changes of connectivity in specific regions as a consequence of social bonding.

## Correlation of brain functional connectivity with partner preference

While consistent longitudinal changes in connectivity were present in our data, we wanted to explore whether some of these nodes had a relationship with the level of partner preference, which captures the strength of the pair-bond (*Williams et al., 1992*), which was evaluated between 48 and 72 hr after the onset of cohabitation.

Positively, our results highlight the role of functional circuits in individual variability of social behavior, in which regions previously associated with reward processing are most relevant in

determining bonding behavior. These correlations are dynamic in time, and each may contribute to the encoding and behavioral outcome of the pair-bond.

LS–VP baseline connectivity positively predicted the level of partner preference. It has been reported that AVP receptor overexpression in VP induces partner preference in socially promiscuous meadow voles (*Lim et al., 2004*). Twenty-four hours after the onset of cohabitation, a negative correlation was found between BLA–NAcc connectivity and partner preference. Variation in OXT receptor density in the NAcc relates to partner preference in prairie voles (*King et al., 2016*). Two weeks after the onset of cohabitation, LS–RSC negatively correlated with partner preference. Previous work demonstrated that AVP receptor expression in the RSC predicts sexual fidelity and territorial behavior in male prairie voles (*Ophir et al., 2008*) and may reflect different mating tactics, that is 'wanderer' or 'resident', in which resident voles maximize mating success through mate guarding and reduced territory space, in contrast to wanderers that overlap many home ranges to increase opportunistic mating (*Getz et al., 2005*). Our findings suggest that the baseline state of reward circuits (LS–VP) may influence initial bond development that will be encoded and established by BLA–NAcc, but long-term pair-bond maintenance may be modulated by social-reward, context-related regions (LS–RSC). In general, network functional connectivity variability may also reflect behavioral diversity influenced by genetic and environmental factors.

The prairie vole has proven to be a valuable model that has enabled the characterization of the neurobiological mechanisms behind complex socio-sexual behaviors, potentially useful to understand human social bonding and its alterations in psychological disorders. To our knowledge, this is the first study to demonstrate correlations between social bonding with brain functional connectivity, as well as time-dependent changes of multiple interacting brain regions (networks) of male and female prairie voles in cohabitation with mating. However, there are some limitations related to the use of rsfMRI. First, the animals were scanned under anesthesia, which may potentially alter the brain functional connectivity. Yet, we have used an anesthesia protocol that practically preserves the functional interactions as in the awake state (*Grandjean et al., 2014*; *Paasonen et al., 2018*) and that has not been reported to significantly alter rsfMRI and BOLD response in longitudinal sessions (*Adamczak et al., 2010*; *Weber et al., 2006*; *Zhao et al., 2008*). Although functional neuroimaging of awake prairie voles is possible, the acclimation process may induce significant stress and even increase the risk of physical harm (*Yee et al., 2016*), limiting the longitudinal design and the interpretation of the results.

Second, unlike male voles, female voles were subjected to surgical procedures and hormonal treatment before the MR scanning sessions. Although male voles are not precise controls for females, no significant differences were identified between male and female voles, behavior-wise or in rsfMRI connectivity. Also, since no significant longitudinal changes in functional connectivity were found in brain structures not expected to be involved in social behavior, the observed changes may be mostly related to social bonding and behavior. However, a control group with no behavioral protocols and same hormonal conditions as the ones reported in this study is desirable to confirm that there are no confounding effects in the longitudinal analysis.

Third, the precision in the definition of the ROIs is limited by the shape and the size of the voxels, being hard to assure that the defined regions specifically and uniquely include signal from the anatomical ROIs. Nevertheless, we are certain that the defined ROIs include the actual ROIs, and at the least, the changes here reported are related to those areas and their surrounding tissue. It is important to note that functionally connected nodes may not necessarily have direct axonal projections to each other, and the interaction between nodes could be mediated or relayed through other structures (*Friston et al., 1993*). Also, contrary to other electrophysiological or neuropharmacological methods, functional MRI captures an indirect measure of neuronal activity (*Kim and Ogawa, 2012*). However, it poses the great advantage that it allows the longitudinal exploration of the brain, with the best spatial resolution and wider coverage that any imaging method can achieve non-invasively. The consistency of our results with recent findings using direct electrophysiology readings (*Amadei et al., 2017*) further supports their relevance in identifying the neurophysiology of complex social behaviors. With evidence from previous reports as support, ROIs analyzed in this study constitute at least three different networks and have multiple roles in the development, establishment, and maintenance of social bonding. Additional research is required to increase understanding on their precise role in prairie vole behavior.

In conclusion, our findings suggest the existence of several functional brain networks involved in the modulation of social bonding in the prairie vole. Network-based statistics revealed a network involving cortical and striatal regions in which its functional connectivity predicted the onset of affiliative behavior, and another overlapping network was associated with the amount of social interaction after pair-bonding. Furthermore, a network with significant changes in time was revealed, with the lateral septum playing a role as a central hub that connects prefrontal and cortical regions with regions from the limbic system and the ventral striatum. Additionally, the retrosplenial cortex-lateral septum, lateral septum-ventral pallidum, and basolateral amygdala-nucleus accumbens functional connectivity correlate with the level of partner preference. In summary, brain functional connectivity allows exploring the mechanisms that underlie individual variability in the expression of socio-sexual behavior, enabling its prediction, and sexual experience and long-term cohabitation induce network-wide changes in socio-sexual relevant circuits. Overall, these results provide a novel approach and analyses to further investigate the neurophysiology of complex social behaviors displayed in the prairie vole.

# Materials and methods

## Key resources table

| Reagent type (species) or resource | Designation | Source or reference | Identifiers | Additional information |
|---|---|---|---|---|
| Software, algorithm | Network-Based Statistic Toolbox | PMID:20600983 | RRID:SCR_002454 | |
| Software, algorithm | Network-Based R-Statistics | doi: 10.1101/2020.11.07.373019 | RRID:SCR_019114 | |
| Software, algorithm | nlme | CRAN | RRID:SCR_015655 | |
| Software, algorithm | FSL | PMID:21979382 | RRID:SCR_002823 | |
| Software, algorithm | ANTS – Advanced Normalization ToolS | PMID:20851191 | RRID:SCR_004757 | |
| Software, algorithm | ggplot2 | CRAN | RRID:SCR_014601 | |
| Software, algorithm | GraphPad Prism | GraphPad | RRID:SCR_002798 | |
| Software, algorithm | MATLAB | MathWorks | RRID:SCR_001622 | |
| Chemical compound, drug | Dexmedetomidine | Zoetis | PubChem CID: 5311068 | s.c. bolus dose: 0.05 mg/kg |
| Chemical compound, drug | Isoflurane | PiSA | PubChem CID: 3763 | |
| Chemical compound, drug | β-Estradiol 3-benzoate | Sigma | PubChem CID: 222757 | Dose: 0.5 mg/mL dissolved in corn oil |
| Other | Dow Corning Silastic Laboratory Tubing | ThermoFisher Scientific | | Length of tubing for capsule: 1.7 cm |

## Animals

Thirty-two 3-month-old sexually naïve female ($N$ = 16) and male ($N$ = 16) prairie voles (*M. ochrogaster*) were used in the study. The animals were housed in a temperature (23°C) and light (14:10 light–dark cycle) controlled room and provided with rabbit diet HF-5326 (LabDiet, St. Louis, MO) oat, sunflower seeds, and water ad libitum. These voles were previously weaned at 21 days, housed in same-sex cages, and were descendants of voles generously donated by Dr. Larry J. Young from his colony at Emory University. The number of subjects per group is comparable to the largest sample sizes using rsfMRI in rodents (*Bajic et al., 2016*; *Christiaen et al., 2019*; *Grandjean et al., 2014*). All surgical, experimental, and maintenance procedures were carried out in accordance with the 'Reglamento de la Ley General de Salud en Materia de Investigación para la Salud' (Health General Law on Health Research Regulation) of the Mexican Health Ministry that follows the National Institutes of Health's 'Guide for the Care and Use of Laboratory Animals' (NIH Publications No. 8023, revised 1978). The animal research protocols were approved by the bioethics committee of the Instituto de Neurobiología, UNAM.

## Surgical procedures

Fourteen days before the experimental protocol, female voles were bilaterally ovariectomized. After recovery, silastic capsules (Dow Corning Silastic Laboratory Tubing; ThermoFisher Scientific, Pittsburg, PA) containing estradiol benzoate (E2B; Sigma–Aldrich, St. Louis, MO) dissolved in corn oil (0.5 mg/mL of E2B) were implanted via s.c. to induce sexual receptivity 4 days before cohabitation protocol and remained implanted during the entire experimental protocol. Females in Microtus species are induced ovulators and do not show cyclic changes (*Taylor et al., 1992*), and for the purpose of this experiment, estrogen-in-oil capsule implantation allowed a stable and equal hormonal dose in all female subjects. The corresponding dose and procedure reliably induced sexual receptivity (*Ingberg et al., 2012*).

## Anesthesia for image acquisition

Animals were anesthetized to avoid stress and excessive movement during scanning sessions. Isoflurane at 3% concentration in an oxygen mixture was used for induction and positioning in the scanner bed, in which the head was immobilized with a bite bar and the coil head holder. Once voles were securely placed in the scanner bed, isoflurane anesthesia (Sofloran; PiSA, Mexico) was adjusted at a 2% concentration and a single bolus of 0.05 mg/kg of dexmedetomidine (Dexdomitor; Zoetis, Mexico) was administered subcutaneously. Five minutes after the bolus injection, isoflurane anesthesia was lowered and maintained at 0.5%. MRI acquisition started when physiological readings were stabilized (~15 min after bolus injection). The use of both anesthetics has been reported as optimal for rsfMRI acquisition in rodents and closely resemble an awake condition (*Grandjean et al., 2014*; *Paasonen et al., 2018*), yet the mentioned combination of dose and administration route were previously standardized for this specific protocol in prairie voles (unpublished data). Body temperature was maintained with a circulating water heating pad within the scanner bed, respiration rate was monitored with an MR-compatible pneumatic pillow sensor, and blood oxygen saturation was measured with an MR-compatible infrared pulse-oximeter (SA Instruments Inc, Stony Brook, NY). After the scanning sessions, animals were monitored until fully recovered and transferred back to their housing.

## Image acquisition

MRI acquisition was conducted with a Bruker Pharmascan 70/16US, 7 Tesla magnetic resonance scanner (Bruker, Ettlingen, Germany), using an MRI CryoProbe transmit/receive surface coil (Bruker, Ettlingen, Germany). Paravision-6 (Bruker, Ettlingen, Germany) was used to perform all imaging protocols. Before running the fMRI sequence, local field homogeneity was optimized within an ellipsoid covering the whole brain and skull using previously acquired field maps. rsfMRI was acquired using a spin-echo echo-planar imaging (SE-EPI) sequence: repetition time (TR) = 2000 ms, echo time (TE) = 19 ms, flip angle (FA) = 90°, field of view (FOV) = 18 × 16 mm$^2$, matrix dimensions = 108 × 96, yielding an in-plane voxel dimensions of 0.167 × 0.167 mm$^2$, and slice thickness of 0.7 mm, total volumes acquired = 305 (10 min and 10 s). EPI bandwidth was 288,461.5 Hz, the number of slices was 25, pi pulse (refocusing pulse) duration was 1.5455 ms with 2200.0 Hz bandwidth, and pi/2 (excitation pulse) was 1.9091 ms and 2200.0 Hz, respectively. After the rsfMRI sequence, an anatomical scan was obtained using a spin-echo rapid acquisition with refocused echoes (Turbo-RARE) sequence with the following parameters: TR = 1800 ms, TE = 38 ms, RARE factor = 16, number of averages (NA) = 2, FOV = 18 × 20 mm$^2$, matrix dimensions = 144 × 160, slice thickness = 0.125 mm, resulting in isometric voxels of size 0.125 × 0.125 × 0.125 mm$^3$.

## Cohabitation and behavior analysis

Female and male voles unrelated to each other were randomly assigned as couples and placed for cohabitation in a new home cage with fresh bedding to promote ad libitum mating and social interaction. The first 6 hr of cohabitation was video recorded for subsequent analysis of social and mating behavior: mount, intromission, and ejaculation latencies from male voles; lordosis latency from females; and huddling latencies for each male and female pair. In male voles, mount latency was scored if it straddled the female from behind with pelvic thrusting, a continued mount behavior with repetitive thrusting was scored as intromission latency, and ejaculation latency was scored if after intromission and deeper thrusting an evident period of male inactivity was followed. Lordosis latency

was scored if the female vole adopted an immobile posture with concave back flexion, neck extension, elevation of the hindquarters, and tail deviation to facilitate male mounting and intromission. Some females displayed lordosis reflex as pre-mounting behavior and were scored for lordosis latency. Since the onset of behaviors such as grooming or licking involve social contact and may overlap with huddling (*Burkett et al., 2016*), in this study huddling latency was only scored if there was continuous, side-to-side bodily contact for at least 10 s. Once joined, voles were housed in couples for the remaining of the experiment and were only separated for MRI scanning sessions and behavioral tests.

## Partner preference test

Subjects underwent a 3 hr PPT to evaluate pair-bond formation. This protocol was based on a previously described test (*Williams et al., 1992*) and was performed in custom-built, three-chambered clear plastic arenas divided by perforated clear plastic barriers that allowed visual, auditory, and olfactory contact, but not physical interaction or mating behavior between subjects. In the central chamber, the vole being tested could roam freely, and time spent in the incentive areas at opposite sides of the chamber was recorded. The incentive areas were defined as the proximal space next to the chambers with its 'partner' or with an opposite-sex, novel 'stranger' vole. All stranger voles were unrelated to subjects in the test and had the same age and hormonal condition than the sexual partner. On each test, partner or stranger voles were randomly and alternately positioned on the opposite chambers of the arena. Each subject was tested only once in the PPT, either at 48 hr or at 72 hr after the onset of cohabitation. If a vole had PPT (assigned randomly and alternately) at the 48 hr period, its partner was tested at the 72 hr period to enable rest between tests and avoid excessive stress. PPT data analysis was performed with *UMATracker* software (*Yamanaka and Takeuchi, 2018*), which allowed quantification of the proportion of time spent with each of the stimulus voles. The percentage of time spent on the area related to the partner and of the area of the stranger were obtained. From this data, a partner preference index was calculated for each subject, consisting of the proportion of time spent on the area related to the partner divided by the proportion of time spent on both social incentive areas (time with partner plus time with stranger vole).

## Imaging data pre-processing

Imaging data pre-processing was performed with FMRIB's Software Libraries (FSL; *Jenkinson et al., 2012*) and Advanced Normalization Tools (ANTs; *Avants et al., 2011*) for spatial registration. To avoid initial signal instability, the first five volumes of each functional series were discarded. The slice-timing correction and motion correction were applied, using the first non-discarded volume as reference. The reference volume was also taken to determine the deformable transformation to the corresponding anatomical image. The resulting transformation was combined with a non-linear transformation to a prairie vole brain template obtained from previously published work (*Ortiz et al., 2018*; *Figure 1—figure supplement 3*). Functional images were later warped to the brain template and resampled to a resolution of 0.16 × 0.16 × 0.7 mm$^3$. To minimize physiological confounds, the first five eigenvectors (time series) within the combined non-gray matter mask were obtained (*Behzadi et al., 2007*), since recent findings have shown that vascular, ventricle, and white matter signal regression enhances functional connectivity specificity from rsfMRI data (*Grandjean et al., 2020*). These eigenvectors and the six motion parameters (three rotations, three displacements) were regressed out from each subject's functional series. Datasets were band-pass filtered to retain frequencies between 0.01 and 0.1 Hz (*Gorges et al., 2017*). Finally, smoothing was applied with a box kernel with size of three voxels, using FSL.

## Functional connectivity analysis

Regions of interest (ROIs) were manually defined on the anatomical prairie vole brain template, visually guided with the Allen Mouse Brain Atlas (*Lein et al., 2007*), to select the minimum possible number of voxels that included/covered the respective region. Connectivity matrices were calculated using MATLAB (Mathworks, Natick, MA) as follows: first, the average time series for each region were extracted from the pre-processed fMRI datasets and then, partial-correlation estimates were obtained for all possible pairs of ROIs, partialing out the remaining time series. Before analysis,

correlation values were Fisher z-transformed. For each region, seed-based functional connectivity maps (*Figure 1—figure supplement 4*), as well as signal-to-noise ratio values, were generated for the assessment of data quality (*Figure 1—figure supplement 5C*).

Due to technical problems, two subjects missed the baseline MRI acquisition (session 1), and two subjects missed the 2 week cohabitation MRI acquisition (session 3). Additionally, two baseline datasets showed signal loss in the dorsal cortex (*Figure 1—figure supplement 5*). The final imaging sample consisted of 90 datasets, with only six subjects missing one session.

Correlations between behavioral data and functional connectivity were assessed with the NBS framework, using the NBS Toolbox (*Zalesky et al., 2010*), which estimates the statistical significance of clusters of connections (networks) by comparing their strength with a null distribution of such property obtained with 5000 permutations of the original data. Since the NBS Toolbox is restricted to general linear hypothesis testing, the longitudinal data was analyzed with linear mixed models (LMM), implemented via the Network-Based R-statistics package (NBR; *Gracia-Tabuenca and Alcauter, 2020*), which allows to implement LMM in an NBS framework. Also, NBR restricts the permutation of within-subject data based on the random variables. Specifically for functional connectivity longitudinal analysis, LMM were fitted for each connection (edge) considering sex (male, female) as a fixed variable, and session (baseline, 24 hr, and 2 weeks of cohabitation) and intercept as random variables. Those edges with significant (p<0.05) sex, session, or interaction effects at this level were used to identify sets of connected nodes, that is components or networks, and their statistical strength (sum of their statistical estimates, Z-values) were compared with a null distribution of the largest clusters from the statistical tests of the permutated data (5000 permutations), this being the NBS framework (*Zalesky et al., 2010*).

## Group-independent component analysis

gICA was applied using FSL's melodic (FSL; *Jenkinson et al., 2012*) on the same pre-processed images previously used in this study, only the female and male prairie voles that underwent all three MR scanning sessions were included on the analysis (n = 26), each subject having three different pre-processed images (n = 78). The number of components was set to 10, as described in previous work in prairie voles by our own group (*Ortiz et al., 2018*). To identify connected voxels over background, gICA maps were scaled to Z-scores and thresholded voxel-wise at Z $\geq$ 2.3 based on a Gaussian/Gamma mixture model and an alternative hypothesis testing approach. These gICA maps were visually inspected and labeled according to their anatomical distribution and location of their maximal regions, with additional visual guidance from the Allen Mouse Brain Atlas (*Lein et al., 2007*).

## Statistical analysis

Data in the study is presented as *mean ± standard error of the mean* unless otherwise noted. Partner preference was explored with one-tailed Mann–Whitney U tests given that Shapiro–Wilk normality tests revealed that such data was not normally distributed. Behavioral data was analyzed with Graph-Pad Prism 5 (*GraphPad* Software, La Jolla, CA).

The NBS Toolbox was used with a level of significance of p<0.05 to identify networks (sets of connections) with significant linear associations with behavioral data. Correlation matrices were not thresholded for NBS analyses, but a fixed threshold of T > 1.7 was used to define the connections with significant associations, for both the actual test and the corresponding tests for the permuted data (5000 permutations). A posteriori Pearson's two-tailed correlation tests were obtained for each edge within the significant networks and were depicted with R software package ggplot2 (ggplot2; *Wickham, 2016*) to better describe the relationships between specific connections and the associated behavior.

Implementation of LMM in NBR relies on the 'nlme' R package (nlme; *Pinheiro et al., 2017*) to fit LMM at the edge level and integrate them with the permutation tests required for the NBS framework. Since the degrees of freedom in LMM may vary for each variable (e.g., $F_{(1,30)}$ for Sex vs $F_{(2,56)}$ for Session), the edge-wise threshold was defined at p<0.05. Similar to the GLM-based NBS, no initial threshold was applied to the connectivity matrices prior to LMM-based NBS. To better describe the differences between sessions for each edge in the significant

networks, estimated marginal means post hoc comparisons were used for LMM and were corrected for an FDR q < 0.05.

In order to evaluate whether networks found through gICA maps had significant changes between sessions, dual regression was applied on the spatial maps from the group-average analysis to generate subject-specific associated time series and the corresponding spatial maps (*Beckmann et al., 2009*; *Nickerson et al., 2017*). First, for each subject, the group-average set of spatial maps was regressed into the subject's 4D space-time dataset, obtaining a set of subject-specific time series, one per group-level spatial map. Next, those time series were regressed into the same 4D dataset, resulting in a set of subject-specific spatial maps, one per group-level spatial map (n = 10). Group differences across sessions were assessed using FSL's randomize permutation-testing tool (*Winkler et al., 2014*), in which two-sample paired t-tests were used (session 1 vs session 2, session 1 vs session 3, and session 2 vs session 3, respectively), given that an adequate anova version is not available in randomize for this experimental design.

## Acknowledgements

We thank Dr. Fernando A Barrios for his valuable comments to the manuscript, and Deisy Gasca, Martín García, Alejandra Castilla, Leopoldo González Santos, Nuri Aranda López, and Ma. De Lourdes Lara for their technical assistance. The authors thankfully acknowledge the imaging resources and support provided by the Laboratorio Nacional de Imagenología por Resonancia Magnética (LANIREM), part of the CONACYT's network of national laboratories. This research was supported by grants CONACYT 252756 to WP, 253631 to RGP, UNAM-DGAPA-PAPIIT IN202818 and IN208221 to WP, IN212219-3 to SA, IN203518-3 to RGP, INPER 2018-1-163 and 3230-21216-05-15 to NFD. LJY's contribution was supported by NIH grants P50MH100023 to LJY and P51OD011132 to YNPRC. MF López-Gutiérrez was supported by a fellowship from CONACYT (fellowship #626152) for her Masters of Science studies in Neurobiology at UNAM (Maestría en Ciencias [Neurobiología], UNAM), is a doctoral student from the Programa de Doctorado en Ciencias Biomédicas, Universidad Nacional Autónoma de México (UNAM), and has received CONACyT fellowship (2020-000026-02NACF-17340, cvu600922).

## Additional information

### Funding

| Funder | Grant reference number | Author |
| --- | --- | --- |
| Consejo Nacional de Ciencia y Tecnología | 252756 | Wendy Portillo |
| Consejo Nacional de Ciencia y Tecnología | 253631 | Raúl G Paredes |
| Dirección General de Asuntos del Personal Académico, Universidad Nacional Autónoma de México | IN202818 | Wendy Portillo |
| Dirección General de Asuntos del Personal Académico, Universidad Nacional Autónoma de México | IN212219-3 | Sarael Alcauter |
| Dirección General de Asuntos del Personal Académico, Universidad Nacional Autónoma de México | IN203518-3 | Raúl G Paredes |
| Instituto Nacional de Perinatología | 2018-1-163 | Néstor F Díaz |
| National Institutes of Health | P50MH100023 | Larry J Young |
| Consejo Nacional de Ciencia y Tecnología | 626152 | M Fernanda López-Gutiérrez |

| National Institutes of Health | P51OD011132 | Larry J Young |
| Consejo Nacional de Ciencia y Tecnología | 600922 | M Fernanda López-Gutiérrez |
| Instituto Nacional de Perinatología | 3230-21216-05-15 | Néstor F Díaz |
| Dirección General de Asuntos del Personal Académico, Universidad Nacional Autónoma de México | IN208221 | Wendy Portillo |

The funders had no role in study design, data collection and interpretation, or the decision to submit the work for publication.

## Author contributions

M Fernanda López-Gutiérrez, Data curation, Formal analysis, Investigation, Visualization, Methodology, Writing - original draft, Project administration; Zeus Gracia-Tabuenca, Software, Formal analysis, Visualization, Methodology; Juan J Ortiz, Francisco J Camacho, Supervision, Methodology; Larry J Young, Néstor F Díaz, Resources, Funding acquisition, Writing - review and editing; Raúl G Paredes, Resources, Validation, Writing - review and editing; Wendy Portillo, Conceptualization, Resources, Data curation, Formal analysis, Supervision, Funding acquisition, Validation, Investigation, Methodology, Writing - original draft, Project administration, Writing - review and editing; Sarael Alcauter, Conceptualization, Resources, Data curation, Software, Formal analysis, Supervision, Funding acquisition, Validation, Investigation, Visualization, Methodology, Writing - original draft, Project administration, Writing - review and editing

## Author ORCIDs

M Fernanda López-Gutiérrez ⓘD https://orcid.org/0000-0002-9513-5607
Sarael Alcauter ⓘD https://orcid.org/0000-0001-8182-6370

## Ethics

Animal experimentation: All surgical, experimental and maintenance procedures were carried out in accordance with the "Reglamento de la Ley General de Salud en Materia de Investigación para la Salud" (Health General Law on Health Research Regulation) of the Mexican Health Ministry which follows the National Institutes of Health's "Guide for the Care and Use of Laboratory Animals" (NIH Publications No. 8023, revised 1978). The animal research protocols were approved by the bioethics committee of the Instituto de Neurobiología, UNAM (Protocol 072). All fMRI scanning sessions were performed under a mixture of isoflurane and dexmedetomidine anesthesia, and all surgery was performed under sevoflurane or a mixture of ketamine/xylazine/saline anesthesia, with every effort to minimize suffering.

## Decision letter and Author response

Decision letter https://doi.org/10.7554/eLife.55081.sa1
Author response https://doi.org/10.7554/eLife.55081.sa2

# Additional files

## Supplementary files

• Transparent reporting form

## Data availability

Data generated or analysed during this study are included in the manuscript and supporting files. Source data files have been provided for Figures 3 to 6 and supplementary source data has been provided for Figure 2. Code for Figure 5 is an R-based package available at https://cran.r-project.org/web/packages/NBR/index.html.

The following dataset was generated:

| Author(s) | Year | Dataset title | Dataset URL | Database and Identifier |
|---|---|---|---|---|
| Alcauter S, López-Gutiérrez MF, Gracia-Tabuenca Z, Ortiz JJ, Young LJ, Camacho FJ, Paredes RlG, Díaz NsF, Portillo W | 2021 | Data from: Brain functional networks associated with social bonding in monogamous voles | https://doi.org/10.5061/dryad.1rn8pk0q9 | Dryad Digital Repository, 10.5061/dryad.1rn8pk0q9 |

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
