## [Decision Letter]

**Acceptance summary:**

This work provides new understanding of the changes in brain functional connectivity that occur following pair bonding in the monogamous prairie vole, by analyzing resting state functional MRI (rsfMRI) before and after formation of pair bonding. The authors found that in both males and females partners there is a connectivity change in specific brain networks associated with pair bonding formation. Moreover, it was found that partner preference evaluated at 48 or 72 hours of cohabitation, may predict long-term functional brain connectivity.

**Decision letter after peer review:**

Thank you for submitting your article "Brain functional connectivity modulates social bonding in monogamous voles" for consideration by *eLife*. Your article has been reviewed by three peer reviewers, and the evaluation has been overseen by a Reviewing Editor and Catherine Dulac as the Senior Editor. The following individuals involved in review of your submission have agreed to reveal their identity: Noam Shemesh (Reviewer #1); William Kenkel (Reviewer #2).

The reviewers have discussed the reviews with one another and the Reviewing Editor has drafted this decision to help you prepare a revised submission.

Summary:

The study examines how functional connectivity in the brain of prairie voles changes as a consequence of the formation of pair bonding by analyzing resting state functional MRI (rsfMRI) in anesthetized voles before and after formation of pair bonding. The authors found that in both males and females partners there is a connectivity changes in specific brain networks (including the ACC, VTA, LS and the hippocampus) associate with pair bonding formation. Moreover, they found that partner preference evaluated at 48 or 72 hours of cohabitation, predicted long-term functional connectivity between the medial amygdala and the ventral pallidum.

All reviewers agreed that the findings are interesting and novel. Reviewer #1 noted that "The authors went to great length to scan a relatively large number of animals and also did a very good job with analyses of resting-state data, which can be tricky at times."

Few important concerns were raised which require to be dealt in the revised submission. Generally the three reviewers agreed that the essential revision could be dealt with in the text.

The revisions should include additional analyses, greater level of methodological detail, additional clarity and explanation of the measured behaviors and statistics, thorough discussion to address the issue of lack of control for the temporal dimension or other control groups. Also, all reviewers agreed that the title should be revised as it is misleading.

The reviews are included in their entirety below.

Reviewer #1:

My main major comments regard (1) the lack of controls for the temporal dimension; and (2) the lack of figures showcasing the quality of the data acquired (and potential processing pipeline effects). These could probably be addressed without any specific new experiments but would require quite a bit of new analyses.

1) Control for the 2w temporal dimension

a) While I understand that the surgery and capsule implantation are commonplace for behavioral or neurophysiological experiments, I am worried that they may introduce some temporal confounds that – to my understanding of the Materials and methods – have not been accounted for. How do we know that the some of the changes in female rsfMRI patterns do not covary with recovery from surgery / hormonal cycles? In an ideal scenario, a control group with no behavior would have been added. However, this can also be mentioned as a confounding factor in the Discussion, and the authors could probably reason against this possibility using prior neurophysiological data.

b) Along the same line, the authors should also present data from ROIs that are NOT expected to be related with social behavior, to control for some of these potential confounders.

2) Quality of experiments.

While again commending the authors for the large number of animals scanned for this study, I would need to see figures with raw data to estimate the levels of artifacts in the original images and how they were addressed. For example, were ghosts detected along the time series? How were these dealt with?

To assess all these, I'd expect the authors to include figures with (a) representative raw data from the rsfMRI scans (to show the quality of the raw data arising from the cryogenic coil); (b) time series analyses in ROIs before/after preprocessing; (c) corresponding maps of rs connectivity, along with connectivity matrices for the different brain areas for the different conditions.

These could be presented as supplementary data, although for me it makes sense to include in the main document.

Reviewer #2:

1) Can the choice of huddling latency be justified in more detail? I don't know whether a reader would appreciate the meaning of this measure.

2) On a similar note: partner preference indexes, being calculated by dividing the time spent huddling with the Partner over the total time spent huddling, seem like they might miss out on differences in total social contact, which could be interesting. That is, a vole with 5 seconds spent huddling with its Partner and 10 seconds total huddling would have an equivalent preference index (0.5) as a vole that spent 5 minutes huddling with its Partner and 10 minutes total huddling (0.5), but these would strike me as very different phenotypes. Total time spent huddling could be an interesting variable to consider alongside the latency to huddle and preference index.

3) While I appreciate the careful consideration taken to select ROIs for this study, a strength of whole brain neuroimaging often comes from its ability to survey all brain regions simultaneously and thereby discover regions previously not known to be involved. Social neuroscience has a list of "usual suspects" and it may be the case that other brain regions that have not yet been implicated in pair bonding could still be important. An exploratory analysis could therefore be useful to the field.

4) I appreciated the presentation of the findings. The time-dependent changes, particularly the acute plastic changes were interesting. I also appreciated the figures' presentation of individual data points and longitudinal trends. That said, the effects do not appear very large, since the distributions overlap so much. Of course, there could still be large effects with overlapping distributions if there was just large individual variation that stayed consistent from time point to time point, but that doesn't appear to be the case for most brain regions. Perhaps violin plots would show the distributions and effect sizes more clearly than box plots? This is something I have struggled with in the past as well, and finding a reliable data visualization solution isn't trivial.

5) Subsection “Partner preference measured between 48h and 72h of cohabitation predicts long-term functional connectivity between MeA and VP” – What was the strength of the correlation between partner preference and MeA-VP connectivity at two weeks of cohabitation? I think the reader would assume all brain region correlations were examined, but were all time points considered as well? I imagine there would be difficulties considering the large number of tests being run, but it also seems like measures from the baseline scan pre-bonding would be the most relevant?

6) Each partner in a pair was tested as the subject in a PPT. Were their preferences for one another correlated? That is, if a male showed a high preference index for his female, was that reciprocated?

7) The Materials and methods state "correlation values were Fisher's z-transformed" so the reader does not have a sense of how strong the connectivity truly is between two regions, rather, what is shown is how strong the connectivity is between two regions relative to the connectivity between two randomly selected regions. Somewhere the range of correlation strengths should be noted.

This manuscript does an admirable job distilling a large number of measures and results down into manageable take-home messages for the reader.

To my knowledge, there has not been much work done distinguishing the phases of 24 hours post introduction to 2 weeks post introduction. I think the reader would benefit from knowing why this time point was chosen. The subsection “Longitudinal changes in functional connectivity after pair bonding”, mentions mate and territorial guarding but these behaviors would have been present by 24 hours.

8) As shown in Figure 3A, the mPFC showed consistent declines in functional connectivity. Beyond my facetious first reaction that this shows romance does indeed impair higher order cognitive processes, I think this is an interesting finding. However, the authors then state "A long-term increase of connectivity was detected between the LS with the mPFC" which doesn't appear to agree with Figure 3A unless I'm misunderstanding something.

Reviewer #3:

1) The first and most fundamental is that there is no control group, and so interpretation of temporal changes associated with bonding assumes that repeated anesthesia and time had no effect on patterns of functional connectivity. Are there any studies that explicitly address this concern? They cite two studies suggesting their protocol "practically preserves the functional interactions as in the awake state", but this falls short of addressing the possibility that repeated anesthesia might change their measures. I really wish they had allocated some of their effort to examining a suitable control-group (sib-housed, age-matched adults, in which control females were OVX + E implanted).

2) The second is that use of NBS and its integration with the LMM is insufficiently described. I cannot tell how they did their analyses based on the provided description (subsection “Statistical analysis”).

3) Apparently the connectivity matrix that serves as an input to NBS is based on the partial correlation matrix at each time point. What was the threshold the authors used to define connectivity? This is essential for the NBS statistic but I could not find it in the Materials and methods or Results.

4) Subsection “Functional connectivity analysis”, they say that linear mixed models were fit, but they are not explicit about what the dependent variables are for these models. The following sentence says that networks "with significant sex, session or interaction effects were identified based on the Network Based Statistics." It is not at all clear what this means. NBS are tests for longer-than-random spans of significant links. Did they simply use NBS to identify significant spans of linkages, and then test each region that went into a significant span with an LMM? The following sentence makes it sounds as though the connections themselves were defined by significant effects of sex or time, not by correlations. The "Statistics" and "Functional connectivity" subsections of the Materials and methods need to be revised in order to be appropriately evaluated. Please be explicit about the relationship between the LMM and the NBS.

5) If sex and time were part of the NBS analysis, I would also ask the authors to be explicit about how they executed the permutation test of the NBS statistic. I assume they permuted the assignment of both sex and time. Did they preserve the structure of between-individual variation?

6) I have similar concerns about the relationship between NBS and correlations with behavior. I found the methods of analysis hard to follow.

7) I had some concerns about mating and bonding in the study. Regarding the estradiol-in-oil implant method, how long do females remain in behavioral estrus in this condition? Preventing pregnancy and extending estrus may distort the dynamics of bonding.

8) The authors only recorded mating behavior over 6h. On a related note, the strength of partner preference seems surprisingly weak (counting each subject as independent across a total 32 tests, but p only ~0.03-0.04); the authors claim an n=32, but tested 32 subjects twice (48h and 72h). In the subsection “Statistical analysis”, the authors say they used a Mann Whitney U, but in Figure 2, they seem to show results of the 2 partner-preference tests lumped together. The authors should be explicit about this analysis to ensure they have not pseudoreplicated their sample size. (If they included 64 tests from 32 individuals in 16 pairs, that would certainly seem to be a problem with pseudoreplication.) Moreover, some pairs did not mate in the 6 hour window. It seems possible that some of the effects of the 2 week vs. 24h time period might be due to the lack of mating in the initial 24h period. I wonder how the results would change if they excluded animals who didn't mate and/or showed no partner preference?

---

## [Author Response]

Reviewer #1:My main major comments regard (1) the lack of controls for the temporal dimension; and (2) the lack of figures showcasing the quality of the data acquired (and potential processing pipeline effects). These could probably be addressed without any specific new experiments but would require quite a bit of new analyses.1) Control for the 2w temporal dimensiona) While I understand that the surgery and capsule implantation are commonplace for behavioral or neurophysiological experiments, I am worried that they may introduce some temporal confounds that – to my understanding of the Materials and methods – have not been accounted for. How do we know that the some of the changes in female rsfMRI patterns do not covary with recovery from surgery / hormonal cycles? In an ideal scenario, a control group with no behavior would have been added. However, this can also be mentioned as a confounding factor in the Discussion, and the authors could probably reason against this possibility using prior neurophysiological data.

We thank the reviewer for pointing out the need to discuss the potential confounds of the capsule implantation. First, we have extended the description of female prairie voles hormonal conditions to highlight the need of such implantation. The following text has been added in the Materials and methods subsection “Surgical procedures”:

“Females in *Microtus* species are induced ovulators and do not show cyclic changes (Taylor et al., 1992), and for the purpose of this experiment, estrogen-in-oil capsule implantation allowed a stable and equal hormonal dose in all female subjects. The corresponding dose and procedure has reliably induced sexual receptivity in rodents (Ingberg et al., 2012).”

Although male voles are not precise controls for females, we did not identify any significant longitudinal changes between females and males, nor in functional connectivity nor in behavioral tests, which could have been potentially attributed to the surgery. As suggested by the reviewer we have included a brief discussion within the limitations of the study, including the reference to the ideal scenario of a control group. The following text is now included:

“Second, unlike male voles, female voles were subjected to surgical procedures and hormonal treatment before the MR scanning sessions. […] However, a control group with no behavioral protocols and same hormonal conditions as the ones reported in this study is desirable to confirm there are no confounding effects in the longitudinal analysis.”

Finally, in order to highlight the relevance of the brain-behavior associations here presented as suggested by the editor, we now present behavioral associations with functional connectivity as the main results, then we show the longitudinal changes, highlighting those with significant associations with social behaviors, and discussing the potential limitations of our findings.

b) Along the same line, the authors should also present data from ROIs that are NOT expected to be related with social behavior, to control for some of these potential confounders.

We appreciate this suggestion. We tested an independent set of brain structures not expected to be involved in social behavior using the same data, i.e., connectivity matrices, with the same analysis methods described previously. As expected, no significant differences were identified.

The following text has been added in the Results section:

“To control for potential confounders, we selected an independent set of brain structures not expected to be involved in social behavior and were tested with the same data and analysis methods than the ROIs mentioned previously. […] Linear mixed models (LMM) implemented via the Network-Based R-statistics package (NBR; Gracia-Tabuenca and Alcauter, 2020) considering sex as a fixed variable, and session and intercept as random variables, found no significant networks with differences between sessions (pFWE=0.1568), nor sex*session interactions (pFWE=0.619 and pFWE=0.3554), suggesting that in both female and male prairie voles, cohabitation with mating and social bonding do not influence changes in functional connectivity in these regions between any session (baseline, 24h and 2 weeks after the onset of cohabitation).”

2) Quality of experiments.While again commending the authors for the large number of animals scanned for this study, I would need to see figures with raw data to estimate the levels of artifacts in the original images and how they were addressed. For example, were ghosts detected along the time series? How were these dealt with?To assess all these, I'd expect the authors to include figures with (a) representative raw data from the rsfMRI scans (to show the quality of the raw data arising from the cryogenic coil); (b) time series analyses in ROIs before/after preprocessing; (c) corresponding maps of rs connectivity, along with connectivity matrices for the different brain areas for the different conditions.These could be presented as supplementary data, although for me it makes sense to include in the main document.

We thank the reviewer for these suggestions. We have included new figure supplements for Figure 1 to keep the main focus of the manuscript on main results, allowing the interested readers to check the data quality.

Reviewer #2:1) Can the choice of huddling latency be justified in more detail? I don't know whether a reader would appreciate the meaning of this measure.

Thank you for the suggestion. We have extended the description of the relevance of this measure in the study of social behaviors. The following text has been added in the Materials and methods subsection “Cohabitation and behavior analysis”:

“The amount and proneness to affiliative behavior has shown reliability in describing the level of sociability and mating systems in Microtus voles. […] Other studies in voles have also measured huddling latency to assess levels of affiliative behavior, since it may influence pair bond induction and formation (Shapiro et al., 1990; Amadei et al., 2017).”

2) On a similar note: partner preference indexes, being calculated by dividing the time spent huddling with the Partner over the total time spent huddling, seem like they might miss out on differences in total social contact, which could be interesting. That is, a vole with 5 seconds spent huddling with its Partner and 10 seconds total huddling would have an equivalent preference index (0.5) as a vole that spent 5 minutes huddling with its Partner and 10 minutes total huddling (0.5), but these would strike me as very different phenotypes. Total time spent huddling could be an interesting variable to consider alongside the latency to huddle and preference index.

This is a very interesting observation. As a reminder, as mentioned in the “Partner Preference Test” subsection in Materials and methods, due to the physical restrictions placed on the chamber, instead of measuring time spent huddling, we measured the percentage of time spent in the incentive areas related to the partner or the stranger vole. Therefore, the partner preference index was calculated by dividing the percentage of time spent on the partner social incentive area over the percentage of time spent on both social incentive areas.

To address the particular question of variability in total social interaction during the test, the proportion of time spent on social incentive areas over the total recorded time was calculated for all subjects. The average percentage of time spent on social incentive areas over the total time of the test was 79.70 ± 2.65 %. Spearman correlation analyses showed that the amount of time spent on social incentive areas had no relationship with the partner preference index (rs(30)=0.08912 p=0.6277), (see new Figure 4—figure supplement 1). However, we tested if functional connectivity 24h after the onset of cohabitation had a relationship with the total time spent on social incentive areas during the test, regardless if it interacted with the partner or stranger prairie vole (*N=32*). Using the NBS toolbox, a significant positive linear relationship (*p* ≤ 0.013) was found with the following network component: LS-NAcc-mPFC-MeA-VP-MOB-DG (Figure 4A).The correlation strength between each connected node was obtained with a posteriori Pearson correlations: LS-NAcc (r(30) = -0.369, *p* = 0.037), NAcc-mPFC (r(30) = -0.312, *p* = 0.081), mPFC-MeA (r(30) = -0.373, *p* = 0.034), MeA-VP (r(30) = -0.376, *p* = 0.033), VP-MOB (r(30) = -0.531, *p* = 0.001), and MOB-DG (r(30) = -0.404, *p* = 0.021) (Figure 4B-G).

This findings suggest that the lower the connectivity between these regions 24h after the onset of cohabitation, the longer the subject would interact with a conspecific of the opposite sex (known or unknown) during the PPT, which was evaluated between 48 and 72h after the onset of cohabitation. Hence, it reflects a phenotype independent both from pre-bonding behavior and partner preference, since regardless of the preference for the partner, prairie voles of both sexes had a differential interest in engaging socially. We have added this new result in the Results section:

“During the partner preference test (PPT), the proportion of time spent on social incentive areas over the total recorded time was calculated for analysis in all subjects (see Materials and methods). […] The correlation strength for each connection in such network was obtained with a posteriori Pearson correlations: LS–NAcc (r(30) = -0.369, p = 0.037), NAcc–mPFC (r(30) = -0.312, p = 0.081), mPFC–MeA (r(30) = -0.373, p = 0.034), MeA–VP (r(30) = -0.376, p = 0.033), VP–MOB (r(30) = -0.531, p = 0.001), and MOB–DG (r(30) = -0.404, p = 0.021) (Figure 4B-G.”

The results are commented in the Discussion section:

**“**A network component detected at 24h after the onset of cohabitation, LS–NAcc–mPFC–MeA–VP–MOB–DG, was negatively correlated with the amount of social incentive a subject would have during the Partner Preference Test, a test in which it interacted with two conspecifics of the opposite sex, one being the partner and the other a stranger vole. […] Apparently, its change in modulation is on a short term since a correlation at 2 weeks of cohabitation was not found and other regions or networks may play this role on a long-term basis.”

3) While I appreciate the careful consideration taken to select ROIs for this study, a strength of whole brain neuroimaging often comes from its ability to survey all brain regions simultaneously and thereby discover regions previously not known to be involved. Social neuroscience has a list of "usual suspects" and it may be the case that other brain regions that have not yet been implicated in pair bonding could still be important. An exploratory analysis could therefore be useful to the field.

Undoubtedly, exploratory analyses are useful for the discovery of regions not previously suspected to be involved in a particular behavior. To address the exploration of large-scale functional brain networks, we have included a gICA in the following sections:

Results section:

“Group independent component analysis (gICA) was also performed to address the exploration of large-scale functional brain networks and potential differences between sessions. […] However, the obtained FWE-Corrected P statistics found no significant differences between sessions. Being this a voxel-wise method, it may not be sensitive enough to detect punctual, region-specific changes in brain functional connectivity.”

Discussion section:

“Group independent component analysis (gICA) was also performed to address the exploration of large-scale functional brain networks and potential differences between sessions. […] Being this a voxel-wise method, it may not be sensitive enough to detect punctual, region-specific changes in brain functional connectivity.”

Materials and methods:

“gICA was applied using FSL’s melodic (FSL; Jenkinson et al., 2012) on the same pre-processed images previously used in this study, only the female and male prairie voles that underwent all 3 MR scanning sessions were included on the analysis (n=26), each subject having 3 different pre-processed images (n=78). […] These gICA maps were visually inspected and labeled according to their anatomical distribution and location of their maximal regions, with additional visual guidance from the Allen Mouse Brain Atlas (Lein et al., 2007).”

Materials and methods subsection “Statistical analysis”:

“In order to evaluate if networks found through gICA maps had significant changes between sessions, dual regression was applied on the spatial maps from the group-average analysis to generate subject-specific associated timeseries and the corresponding spatial maps (Beckmann et al., 2009; Nickerson et al., 2017). […] Group differences across sessions were assessed using FSL's randomise permutation-testing tool (Winkler et al., 2014), in which two-sample paired t-tests were used (session 1 vs. session 2, session 1 vs. session 3, and session 2 vs. session 3, respectively), given that an adequate anova version is not available in randomise for this experimental design.”

4) I appreciated the presentation of the findings. The time-dependent changes, particularly the acute plastic changes were interesting. I also appreciated the figures' presentation of individual data points and longitudinal trends. That said, the effects do not appear very large, since the distributions overlap so much. Of course, there could still be large effects with overlapping distributions if there was just large individual variation that stayed consistent from time point to time point, but that doesn't appear to be the case for most brain regions. Perhaps violin plots would show the distributions and effect sizes more clearly than box plots? This is something I have struggled with in the past as well, and finding a reliable data visualization solution isn't trivial.

Certainly, our data reflects ample individual variability and the effects of these findings may not appear very large. Nonetheless, partial correlation analyses were used to assess significant functional connections between any two regions minimizing the effects of other regions in such relation. This type of analysis has high sensitivity in the detection of true functional connections, as it first regresses out all other nodes before correlating any two nodes (Smith et al., 2013). Hence, our analysis shows the significant functional connections that group-wise, were the strongest and most consistent in spite of underlying individual variability, the latter also allowing us to explain the observed variability in the expression of social behavior.

Following the advice of the reviewer, we are now presenting data of the mentioned figure (now Figure 5) in violin plots.

5) Subsection “Partner preference measured between 48h and 72h of cohabitation predicts long-term functional connectivity between MeA and VP” – What was the strength of the correlation between partner preference and MeA-VP connectivity at two weeks of cohabitation? I think the reader would assume all brain region correlations were examined, but were all time points considered as well? I imagine there would be difficulties considering the large number of tests being run, but it also seems like measures from the baseline scan pre-bonding would be the most relevant?

Under your suggestion, we have decided instead to analyze partner preference correlations with regions that underwent significant longitudinal changes after social bonding to better address such associations. We have added this new result in the Results section:

“Although several regions were found to change significantly after cohabitation and social bonding, we tested if any specific connection of the detected network component (Figure 5A) could have a relationship with the partner preference index obtained between 48 and 72 hours after the onset of cohabitation, which was used to evaluate the level of pair bonding in each subject. […] Also, significant negative relationships were found between the partner preference index and BLA–NAcc functional connectivity at 24h of cohabitation (session 2) (r(30) = -0.464, p = 0.0074), and in LS–RSC functional connectivity at 2 weeks of cohabitation (session 3) (r(28) = -0.437, p = 0.015 ), which may suggest that the lower the connectivity between these regions at the specified time points, the higher the partner preference index (Figure 6B-C).”

The results are further commented in the Discussion section:

“While consistent longitudinal changes in connectivity were present in our data, we wanted to explore if some of these nodes had a relationship with the level of partner preference, which captures the strength of the pair bond (Williams et al., 1992), which was evaluated between 48 and 72 hours after the onset of cohabitation.

[…] In general, network functional connectivity variability may also reflect behavioral diversity influenced by genetic and environmental factors.”

6) Each partner in a pair was tested as the subject in a PPT. Were their preferences for one another correlated? That is, if a male showed a high preference index for his female, was that reciprocated?

This is a very interesting question that we did not address before. Two-tailed Spearman correlation analysis revealed that partner preference was not reciprocated between male and female partners (rs(14)=-0.203, p=0.450), suggesting partner preference may not be majorly influenced by the partner’s perceived socio-sexual behavior (Figure 4—figure supplement 1).

7) The Materials and methods state "correlation values were Fisher's z-transformed" so the reader does not have a sense of how strong the connectivity truly is between two regions, rather, what is shown is how strong the connectivity is between two regions relative to the connectivity between two randomly selected regions. Somewhere the range of correlation strengths should be noted.

We have included the non-Fisher z-transformed partial-correlation connectivity matrices in the new Figure 1—figure supplement 2.

This manuscript does an admirable job distilling a large number of measures and results down into manageable take-home messages for the reader.

Thank you for your kind comment.

To my knowledge, there has not been much work done distinguishing the phases of 24 hours post introduction to 2 weeks post introduction. I think the reader would benefit from knowing why this time point was chosen. The subsection “Longitudinal changes in functional connectivity after pair bonding”, mentions mate and territorial guarding but these behaviors would have been present by 24 hours.

This time period was selected based on previous literature. Specifically, on studies that have shown neurophysiological and behavioral changes occurring by a two-weeks period that may be related to the maintenance of the pair bond. The following text has been added to include such information in the Introduction section:

“While prairie voles of both sexes already display changes in behavior by 24 hours of cohabitation with mating as a result of pair-bonding (Wang and Aragona, 2004; Williams et al., 1992), the NAcc, for example, has substantially more D1-like receptor binding 2 weeks after female exposure, relative to non–pair bonded males (Aragona et al., 2006), a phenomenon that’s also been observed in female voles (Resendez et al., 2016). […] While there is a possibility that these changes in behavior involve the interplay of broad networks, it has not been directly explored in vivo.”

Altogether, such evidence seemed relevant to explore changes in functional connectivity after cohabitation for two weeks.

8) As shown in Figure 3A, the mPFC showed consistent declines in functional connectivity. Beyond my facetious first reaction that this shows romance does indeed impair higher order cognitive processes, I think this is an interesting finding. However, the authors then state "A long-term increase of connectivity was detected between the LS with the mPFC" which doesn't appear to agree with Figure 3A unless I'm misunderstanding something.

This is an interesting observation and potential interpretation. However, between baseline and 24h after the onset of cohabitation, a non-significant decrease of connectivity is shown between LS-mPFC, but it increased significantly at the 2 weeks period (Session 3). Post-hoc analyses showed LS-mPFC connectivity had significantly increased when comparing Baseline with 2 weeks (t(56) = -2.856; p=0.0060), and also when comparing 24h vs. 2 weeks LS-mPFC (t(56) = -4.013; p=0.0001) (Figure 5H).

Reviewer #3:1) The first and most fundamental is that there is no control group, and so interpretation of temporal changes associated with bonding assumes that repeated anesthesia and time had no effect on patterns of functional connectivity. Are there any studies that explicitly address this concern? They cite two studies suggesting their protocol "practically preserves the functional interactions as in the awake state", but this falls short of addressing the possibility that repeated anesthesia might change their measures. I really wish they had allocated some of their effort to examining a suitable control-group (sib-housed, age-matched adults, in which control females were OVX + E implanted).

Indeed, a number of studies have addressed the effects of sedation in fMRI longitudinal studies in rodents, specifically in forepaw stimulation. In rats, three fMRI sessions conducted in 5-day intervals, in which a 0.05 mg/kg medetomidine bolus injection plus a subcutaneous infusion of 0.1 mg/kg/h medetomidine were used, found no significant differences in BOLD signal change between sessions (Weber et al., 2006). In mice, repetitive medetomidine sedation with a dose of 0.3 mg/kg in combination with medetomidine infusion of 0.6 mg/kg/h in a 3 fMRI session protocol was also well tolerated and with reproducible BOLD response in forepaw stimulation (Adamczak et al., 2010).

While these studies suggest medetomidine effectiveness for fMRI longitudinal experiments and for rsfMRI (Zhao et al., 2008), the combined use of isoflurane with medetomidine has shown to have better cortical and partially preserved thalamo-cortical connectivity, having a higher resemblance to the awake condition in comparison to isoflurane and medetomidine alone (Grandjean et al., 2014, Paasonen et al., 2018). Even though a control group would certainly be more suitable for assessing the effects of anesthesia in our study, the relatively low dose used for our rodent model and the evidence shown on previous reports, support that our sedation protocol would not have a significant effect on the longitudinal follow-up. A brief sentence addressing previous studies on anesthesia has been included (subsection “Anesthesia for image acquisition”). In addition, we have now added the longitudinal analysis of regions not expected to be involved in social behavior, which had no significant changes between sessions.

However, as we now state in limitations (Discussion), a control group is desirable to confirm there are no confounding effects in the longitudinal changes two weeks after the onset of cohabitation. Accordingly, and also as suggested by the editor and reviewers, we now present the associations between functional connectivity and behavior as the main results, then we show the longitudinal changes, highlighting those with significant associations with social behaviors, discussing the potential limitations of our findings.

2) The second is that use of NBS and its integration with the LMM is insufficiently described. I cannot tell how they did their analyses based on the provided description (subsection “Statistical analysis”).

Edgewise LMM through NBS framework was reformulated as follows in the Materials and methods – subsection “Functional connectivity analysis”:

“Correlations between behavioral data and functional connectivity were assessed with the Network Based Statistics (NBS) framework, using the Network Based Statistic Toolbox (Zalesky et al., 2010), which estimates the statistical significance of clusters of connections (networks) by comparing their strength with a null distribution of such property obtained with 5000 permutations of the original data. […] Those edges with significant (p < 0.05) sex, session or interaction effects at this level were used to identify sets of connected nodes, i.e. components or networks, and their statistical strength (sum of their statistical estimates, Z-values) were compared with a null distribution of the largest clusters from the statistical tests of the permutated data (5000 permutations), this being the NBS framework (Zalesky et al., 2010).”

3) Apparently the connectivity matrix that serves as an input to NBS is based on the partial correlation matrix at each time point. What was the threshold the authors used to define connectivity? This is essential for the NBS statistic but I could not find it in the Materials and methods or Results.

As stated by Zalesky et al., 2010, one of the advantages of NBS is that no initial threshold is required for the connectivity matrices to be analyzed (although it is possible to do so), however it is critical to define a threshold to define the connections that surpass the statistical tests, which will be used to define the clusters of connections. Accordingly, we did not apply any connectivity threshold to the connectivity matrices, but we did apply a threshold at the sample inference level to define the connections that surpassed the statistical tests, and with these, define the clusters of connections or networks. For the behavioral correlates with NBS, using GLM, we used a threshold of T > 1.7. For the LMM-based NBR, we are able to define such a threshold in terms of the p-value, and it was set at p < 0.05 for each edge. Note that when using LMM the degrees of freedom between variables may vary (e.g., F(1,30) for Sex vs. F(2,56) for Session), for this reason we rely on significance instead of a static F-value in the LMM. Nevertheless, control of the family- wise error rate (FWER) of the defined networks (clusters of connections), is controlled irrespective of the threshold choice by the estimation of the null distributions.

The following information has been now included in the Materials and methods “Statistical analysis” subsection:

“The NBS toolbox was used with a level of significance of p < 0.05 to identify networks (sets of connections) with significant linear associations with behavioral data. […] To better describe the differences between sessions for each edge in the significant networks, estimated marginal means (EMM) post-hoc comparisons were used for LMM and were corrected for a false discovery rate (FDR) q < 0.05.**”**

4) Subsection “Functional connectivity analysis”, they say that linear mixed models were fit, but they are not explicit about what the dependent variables are for these models. The following sentence says that networks "with significant sex, session or interaction effects were identified based on the Network Based Statistics." It is not at all clear what this means. NBS are tests for longer-than-random spans of significant links. Did they simply use NBS to identify significant spans of linkages, and then test each region that went into a significant span with an LMM? The following sentence makes it sounds as though the connections themselves were defined by significant effects of sex or time, not by correlations. The "Statistics" and "Functional connectivity" subsections of the Materials and methods need to be revised in order to be appropriately evaluated. Please be explicit about the relationship between the LMM and the NBS.

The dependent variable of the LMM models was the functional connectivity represented in the connectivity matrices (i.e., partial correlations between ROIs). As the reviewer correctly states, NBS tests for longer-than-random spans of significant links, but first we need to identify those significant links. That is done performing a statistical test in each possible connection (element of the matrix), be it using linear mixed models or the general linear model, as specified in any case. So, NBS first performs the statistical test in every element of the matrix, to identify the weighted or binary size of the clusters of links that surpass the test. Then, for every permutation of the original data, it performs the same tests, and identifies the size of the larger cluster to generate a null distribution. Finally, the size of clusters identified in the actual (real) case are compared with the null distribution of the largest random clusters, and the final family-wise p-value for each of them is estimated. So, depending on the data and the question being asked, the NBS framework can be applied with a variety of statistical tests. Shortly, associations between functional connectivity (from a single session) and behavioral data, are performed with the GLM; longitudinal data is analyzed with LMM. The great advantage of the NBS framework is that not only tests individual connections, but it identifies sets of connections, or networks.

We have reviewed the “Statistics” and “Functional connectivity” subsections that concern the NBS framework as follows:

“Correlations between behavioral data and functional connectivity were assessed with the Network Based Statistic (NBS) framework, using the Network Based Statistic Toolbox (Zalesky et al., 2010), which estimates the statistical significance of clusters of connections (networks) by comparing their strength with a null distribution of such property obtained with 5000 permutations of the original data. […] Those edges with significant (p < 0.05) sex, session or interaction effects at this level were used to identify sets of connected nodes, i.e. components or networks, and their statistical strength (sum of their statistical estimates, Z-values) were compared with a null distribution of the largest clusters from the statistical tests of the permutated data (5000 permutations), this being the NBS framework (Zalesky et al., 2010).”

“Statistical analysis. The NBS toolbox was used with a level of significance of p < 0.05 to identify networks (sets of connections) with significant linear associations with behavioral data. Correlation matrices were not thresholded for NBS analyses, but a fixed threshold of T > 1.7 was used to define the connections with significant associations, for both the actual test and the corresponding tests for the permuted data (5000 permutations). […] To better describe the differences between sessions for each edge in the significant networks, estimated marginal means (EMM) post-hoc comparisons were used for LMM and were corrected for a false discovery rate (FDR) q < 0.05.”

5) If sex and time were part of the NBS analysis, I would also ask the authors to be explicit about how they executed the permutation test of the NBS statistic. I assume they permuted the assignment of both sex and time. Did they preserve the structure of between-individual variation?

That is correct, the labels for sex and session were also permuted when they were included in the model, but preserving the data within-subjects to maintain the data structure.

6) I have similar concerns about the relationship between NBS and correlations with behavior. I found the methods of analysis hard to follow.

NBS takes as input the connectivity matrices from all observations and tests their relationship to linear covariates using a general linear model (GLM). The hypothesis tested at every connection was if connectivity strength was linearly associated with behavioral data. Relying on permutations, the NBS framework identifies clusters of connections for which the null hypothesis was rejected. The NBS enables rejection of the null hypothesis at the network level (i.e., at the level of those clusters), controlling the family-wise error rate (FWER). We have modified the description as referred in the previous points.

7) I had some concerns about mating and bonding in the study. Regarding the estradiol-in-oil implant method, how long do females remain in behavioral estrus in this condition? Preventing pregnancy and extending estrus may distort the dynamics of bonding.

Silastic implants were selected over estradiol s.c. administration because daily injections for two weeks may have been a potential stressor on the subjects and its partners. This method also enables more stable plasma estradiol levels than pellets (Mosquera, Shepherd and Torraro, 2015). A pilot study was performed to determine the efficiency of the estradiol implant and females remain sexually receptive for at least 20 days (non-published data).

Regarding female hormonal condition, *Microtus* species are induced ovulators and do not show cyclic changes. Female prairie voles are sexually inactive in the family nest; and stimuli from the environment, particularly male-related olfactory stimuli, are thought to induce endocrine changes, including increases in serum estrogen levels that promote sexual and behavioral receptivity (Taylor et al., 1992). In the presence of an unrelated male, females usually become sexually receptive at 48 h, will mate for 24h (Carter et al., 1987), and may remain in vaginal estrous for as long as 30 days when housed across a barrier from males (Richmond and Conaway, 1969). Therefore, the estradiol-capsule implantation enabled female sexual receptivity and reduced defensive aggression towards the male partner (Winslow et al., 1993; Insel et al., 1995). A brief sentence has been added to remind the reader the particular estrus condition in female prairie voles, in the Materials and methods subsection “Surgical procedures”:

“Females in *Microtus* species are induced ovulators and do not show cyclic changes (Taylor et al., 1992) , and for the purpose of this experiment, estrogen-in-oil capsule implantation allowed a stable and equal hormonal dose in all female subjects.**”**

Additionally, mating is not essential for partner preference formation in female voles, though it has shown to accelerate bonding. Rather, cohabitation leads to pair bonding, and events associated with mating or estrus induction facilitate the onset of partner preference; and pair bonding may be established with at least 6 hours of cohabitation with mating or 24h of cohabitation without mating (Williams et al., 1992).

8) The authors only recorded mating behavior over 6h. On a related note, the strength of partner preference seems surprisingly weak (counting each subject as independent across a total 32 tests, but p only ~0.03-0.04); the authors claim an n=32, but tested 32 subjects twice (48h and 72h). In the subsection “Statistical analysis”, the authors say they used a Mann Whitney U, but in Figure 2, they seem to show results of the 2 partner-preference tests lumped together. The authors should be explicit about this analysis to ensure they have not pseudoreplicated their sample size. (If they included 64 tests from 32 individuals in 16 pairs, that would certainly seem to be a problem with pseudoreplication.) Moreover, some pairs did not mate in the 6 hour window. It seems possible that some of the effects of the 2 week vs. 24h time period might be due to the lack of mating in the initial 24h period. I wonder how the results would change if they excluded animals who didn't mate and/or showed no partner preference?

Each subject was tested only once in the partner preference test (PPT), within a 48h to 72h time frame. If a vole had PPT (assigned randomly and alternately) at the 48h period, its partner was tested at the 72h period. Specifically, each vole participated in two different PPT tests: once as the tested vole, and once as a stimulus vole in its partner’s PPT. Hence, we obtained a single partner preference measure for each subject. This 48h-72h testing period was chosen due to time constraints and to avoid excessive stress on the animals, which have no access to food and water during the 3-hour duration of the test. For better clarity, we have added the following text in the Materials and methods subsection “Partner Preference Test”:

“Each subject was tested only once in the partner preference test (PPT), either at 48 or 72h after the onset of cohabitation. If a vole underwent PPT (assigned randomly and alternately) at the 48h period, its partner was tested at the 72h period to enable rest between tests and avoid excessive stress.”

Regarding the strength of partner preference, as it has been previously characterized in this rodent model, we did find a statistically significant preference for the partner. However, as discussed in the manuscript, there is also a wide variability in partner preference in this species, which gave us the advantage to explore correlations with functional connectivity.

In respect to mating behavior, as mentioned before, mating is not essential for pair bonding in female voles, but it may facilitate its establishment (Williams et al., 1992). Though it has been reported that males may pair with diestrus females and without copulation (Shapiro and Dewsbury, 1990), mating in male voles has reported to be more relevant in pair bond formation than in females (Winslow et al., 1993; Insel et al., 1995). This trend is not entirely clear in our data: of 5 male voles with a partner preference index lower than 0.5, 2 of them did not mate during the 6 h window. It is important to consider that these subjects may have succeeded in mating both before the 24h scanning session or before the PPT. Further investigation would be necessary to fully understand the effects of mating in male vole pair bonding facilitation and its relationship with changes in brain functional connectivity, since only a small number of subjects were found in this particular circumstance.

References:

Adamczak, J. M., Farr, T. D., Seehafer, J. U., Kalthoff, D., and Hoehn, M. (2010). High field BOLD response to forepaw stimulation in the mouse. NeuroImage, 51(2), 704–712.

Carter, C. S., Witt, D. M., Auksi, T., and Casten, L. (1987). Estrogen and the induction of lordosis in female and male prairie voles (Microtus ochrogaster). Hormones and Behavior, 21(1), 65–73.

Grandjean, J., Canella, C., Anckaerts, C., Ayrancı, G., Bougacha, S., Bienert, T., … Gozzi, A. (2020). Common functional networks in the mouse brain revealed by multi-centre resting-state fMRI analysis. NeuroImage, 205(October 2019).

Insel, Thomas R., Preston, S., and Winslow, J. T. (1995). Mating in the monogamous male: Behavioral consequences. Physiology and Behavior, 57(4), 615–627.

Mosquera, L., Shepherd, L., and Torrado, A. I. (2015). Comparison of Two Methods of Estradiol Replacement: their Physiological and Behavioral Outcomes. Journal of Veterinary Science and Technology, 06(06).

Richmond, M. E., and Conaway, C. H. (1969). Induced ovulation and oestrus in Microtus ochrogaster. Journal of Reproduction and Fertility, Supplement(6), 357–376.

Shapiro, L. E., and Dewsbury, D. A. (1990). Differences in affiliative behavior, pair bonding, and vaginal cytology in two species of vole (Microtus ochrogaster and M. montanus). Journal of Comparative Psychology, 104(3), 268–274.

Smith, S. M., Vidaurre, D., Beckmann, C. F., Glasser, M. F., Jenkinson, M., Miller, K. L., … Van Essen, D. C. (2013). Functional connectomics from resting-state fMRI. Trends in Cognitive Sciences, 17(12), 666–682.

Winslow, J. T., Hastings, N., Carter, C. S., Harbaugh, C. R., and Insel, T. R. (1993). A role for central vasopressin in pair bonding in monogamous prairie voles. Nature, 365(6446), 545–548.